# Adenovirus Remodeling of the Host Proteome and Host Factors Associated with Viral Genomes

Joseph M. Dybas,[a,b,c] Krystal K. Lum,[a,b] Katarzyna Kulej,[a,b] Emigdio D. Reyes,[a,b] Richard Lauman,[d,e] Matthew Charman,[a,b] Caitlin E. Purman,[a,b] Robert T. Steinbock,[b] Nicholas Grams,[b] Alexander M. Price,[a,b] Lydia Mendoza,[b] Benjamin A. Garcia,[d,e] Matthew D. Weitzman[a,b,e]

[a]Division of Protective Immunity and Division of Cancer Pathobiology, The Children's Hospital of Philadelphia, Philadelphia, Pennsylvania, USA
[b]Department of Pathology and Laboratory Medicine, University of Pennsylvania Perelman School of Medicine, Philadelphia, Pennsylvania, USA
[c]Department of Biomedical and Health Informatics, The Children's Hospital of Philadelphia, Philadelphia, Pennsylvania, USA
[d]Department of Biochemistry and Biophysics, University of Pennsylvania Perelman School of Medicine, Philadelphia, Pennsylvania, USA
[e]Penn Epigenetics Institute, University of Pennsylvania Perelman School of Medicine, Philadelphia, Pennsylvania, USA

Joseph M. Dybas and Krystal K. Lum contributed equally to this work. Author order was determined based on respective contributions to experiments, data analysis, and manuscript preparation.

**ABSTRACT** Viral infections are associated with extensive remodeling of the cellular proteome. Viruses encode gene products that manipulate host proteins to redirect cellular processes or subvert antiviral immune responses. Adenovirus (AdV) encodes proteins from the early E4 region which are necessary for productive infection. Some cellular antiviral proteins are known to be targeted by AdV E4 gene products, resulting in their degradation or mislocalization. However, the full repertoire of host proteome changes induced by viral E4 proteins has not been defined. To identify cellular proteins and processes manipulated by viral products, we developed a global, unbiased proteomics approach to analyze changes to the host proteome during infection with adenovirus serotype 5 (Ad5) virus. We used whole-cell proteomics to measure total protein abundances in the proteome during Ad5 infection. Since host antiviral proteins can antagonize viral infection by associating with viral genomes and inhibiting essential viral processes, we used Isolation of Proteins on Nascent DNA (iPOND) proteomics to identify proteins associated with viral genomes during infection with wild-type Ad5 or an E4 mutant virus. By integrating these proteomics data sets, we identified cellular factors that are degraded in an E4-dependent manner or are associated with the viral genome in the absence of E4 proteins. We further show that some identified proteins exert inhibitory effects on Ad5 infection. Our systems-level analysis reveals cellular processes that are manipulated during Ad5 infection and points to host factors counteracted by early viral proteins as they remodel the host proteome to promote efficient infection.

**IMPORTANCE** Viral infections induce myriad changes to the host cell proteome. As viruses harness cellular processes and counteract host defenses, they impact abundance, post-translational modifications, interactions, or localization of cellular proteins. Elucidating the dynamic changes to the cellular proteome during viral replication is integral to understanding how virus-host interactions influence the outcome of infection. Adenovirus encodes early gene products from the E4 genomic region that are known to alter host response pathways and promote replication, but the full extent of proteome modifications they mediate is not known. We used an integrated proteomics approach to quantitate protein abundance and protein associations with viral DNA during virus infection. Systems-level analysis identifies cellular proteins and processes impacted in an E4-dependent manner, suggesting ways that adenovirus counteracts

Address correspondence to Matthew D. Weitzman, weitzmanm@chop.edu.

A systems-level analysis from @WeitzmanLab illustrating how Adenovirus remodels the host proteome. Early viral proteins affect cellular protein abundance and host factor association with viral genomes.

potentially inhibitory host defenses. This study provides a global view of adenovirus-mediated proteome remodeling, which can serve as a model to investigate virus-host interactions of DNA viruses.

**KEYWORDS** adenovirus, ubiquitin, protein-DNA binding, mass spectrometry, proteomics, iPOND, virus-host interactions

Viruses manipulate the host cell proteome during infection to redirect cellular processes required for progression through stages of the viral infectious cycle and to combat antagonism by host defenses (1–3). One hallmark of the cellular response to infection is sensing and direct targeting of viral genomes by antiviral host factors, which in turn must be subverted by viral countermeasures to promote productive infection (4–6). A systematic view of host proteome remodeling during virus infection is critical to developing a comprehensive understanding of virus-host interactions and their impact on infection.

Human adenovirus (AdV) is a nuclear replicating DNA virus associated with the formation of distinct sites of viral transcription and genome replication, known as viral replication centers (VRCs) (7, 8). Cellular and viral proteins essential for viral DNA transcription and replication are localized to VRCs (9). On the other hand, host antiviral factors sense and respond to viral genomes at VRCs and are counteracted by AdV (10). These AdV countermeasures include multiple genes contained within the early transcription unit 4 (E4). Mutations within the E4 region are associated with defects in viral mRNA processing, late viral protein production, viral DNA replication, and host cell shutoff (11–16). Adenovirus serotype 5 (Ad5) encodes at least 6 viral proteins from the E4 region (E4orf1, E4orf2, E4orf3, E4orf4, E4orf6, E4orf6/7). The E4orf3 and E4orf6 proteins counteract host antiviral proteins during productive Ad5 infection, in some cases acting redundantly (10–12, 16, 17). The Ad5 E4orf6 and E1B55K proteins interact and recruit a host cullin5 complex with E3 ubiquitin ligase activity, which redirects cellular ubiquitination to modify host proteins (18, 19). Known substrates of the Ad5 E4orf6-E1B55K E3 ligase complex include host proteins targeted for proteasome-mediated degradation. Several of these targets are involved in the DNA damage response (DDR) and apoptosis, including MRE11, RAD50, NBS1, DNA ligase IV, Bloom helicase, TIP60, and p53 (20–29). E1B55K has also been shown to mediate proteasomal degradation of the transcriptional regulator DAXX independently of E4orf6 (30). In addition, the Ad5 E4orf6-E1B55K complex can also mediate nonproteolytic ubiquitination of host RNA-binding proteins RALY and hnRNPC to inhibit binding to viral RNA and promote efficient viral RNA processing (31). E4orf3 induces degradation or mislocalization of host antiviral proteins including PML bodies, TRIM proteins, and the MRN complex during Ad5 infection (25, 32–39). Interestingly, E4orf3 can also induce proteasome-mediated degradation of cellular proteins GTF2I and TIF1γ/TRIM33, although the mechanism is not known (37–40). A variety of host factors have been identified as the substrates of E4 proteins, including antiviral proteins which target viral DNA. However, the complexities underlying Ad5 remodeling of the host proteome throughout infection have not been fully described, and how these changes impact protein interactions with viral DNA is unknown.

The goal of this study was to employ a systems-level approach to quantitate AdV viral impacts on the host proteome and to identify host responses and their opposing viral countermeasures. We implemented a proteomics strategy upon Ad5 infection that quantitated (i) changes in cellular protein abundance and (ii) proteomes associated with replicating viral genomes (Fig. 1). Specifically, we used a whole-cell proteomics (WCP) approach to measure abundances of host and viral proteins over a time course of Ad5 wild-type (WT) infection. We also adapted an Identification of Proteins On Nascent DNA (iPOND) proteomics approach (41–44), which was previously used to measure abundances of proteins associated with replicating Ad5 genomes (45). In the current study, we used iPOND to compare proteomes enriched on viral genomes during infection with Ad5 WT or an AdΔE4 mutant virus (*dl*1004). *dl*1004 lacks most of the

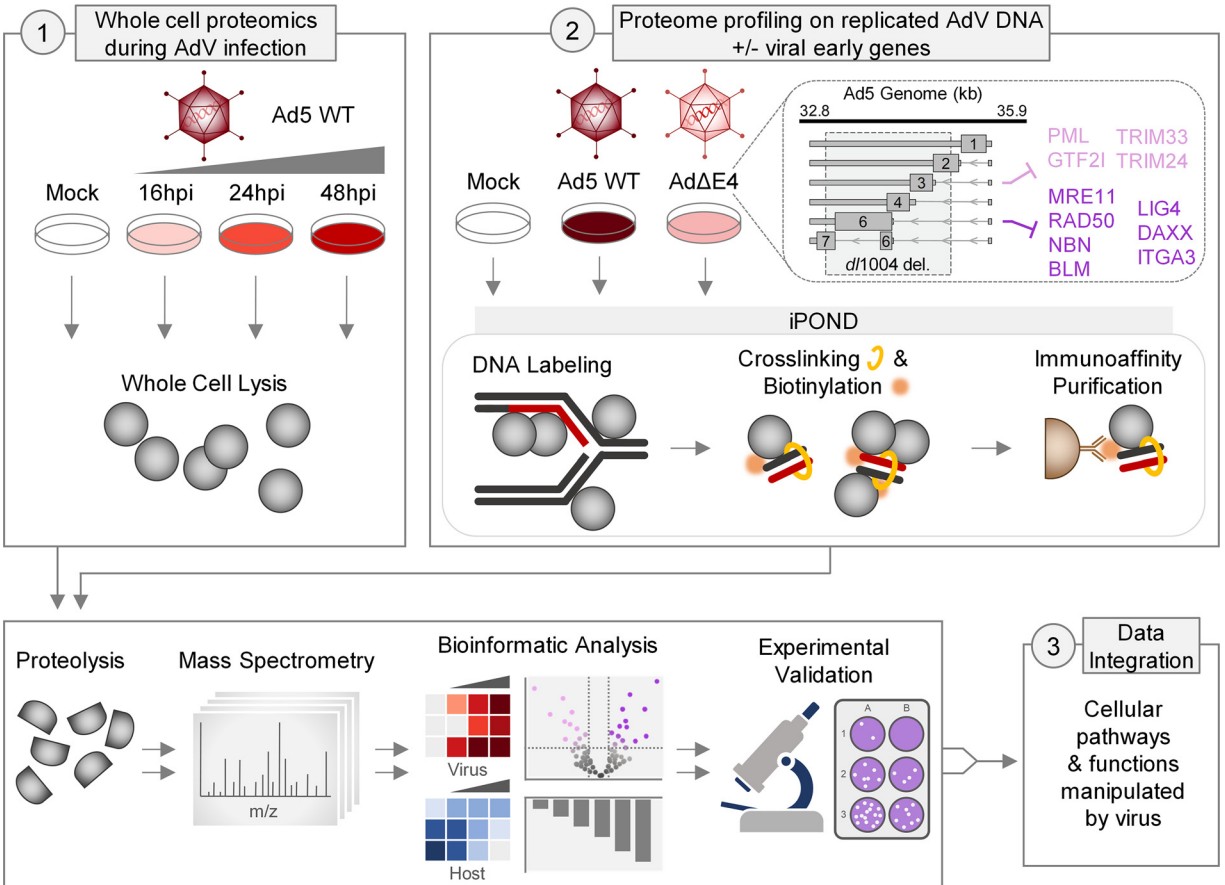

**FIG 1** Platform to define adenovirus manipulation of the host proteome and proteins associated with replicating viral genomes. Proteomic workflow to quantitate proteins within the whole-cell proteome (WCP) and associated with viral genomes by Isolation of Proteins on Nascent DNA (iPOND). Analysis of the WCP for a time course of Ad5 WT infection revealed changes to the host and viral proteome (1). Analysis of the iPOND data comparing Ad5 WT and AdΔE4 infections identified cellular factors associated with replicating viral genomes in the presence and absence of Ad5 early region E4 gene products (2). Schematic of the Ad5 E4 transcription region and known expressed open reading frames (ORFs) is included. The AdΔE4 mutant virus (*dl*1004) lacks ORFs 3, 4, and 6 and portions of Orf2 and Orf6/7. Proteins previously identified as targets of E4orf3 (light purple) and E4orf6 (dark purple) are indicated. After proteolysis and mass spectrometry, proteins were identified and categorized bioinformatically to identify cellular proteins and pathways manipulated during infection. Data integration and experimental validation determined the effects of these pathways on virus infection (3).

E4 region, including E4orf3 and E4orf6 (12). We hypothesized that host proteins associated with AdΔE4 but not Ad5 WT genomes will include antiviral proteins that restrict productive infection and that those antiviral proteins may be targets of E4orf3 and/or E4orf6 countermeasures during Ad5 WT infection.

Our WCP analysis revealed distinct sets of cellular proteins and associated processes which are dynamically altered during Ad5 WT infection. Our iPOND analysis comparing Ad5 WT and AdΔE4 mutant infections revealed distinct proteomes enriched on viral genomes. Upon integration of these data sets, our systems-level approach highlighted cellular processes, such as those involving chromosome organization and genome maintenance, that are potentially manipulated via E4 impacts on the cellular proteome. Our data uncovered proteins from two families, the Switch/Sucrose Non-Fermentable (SWI/SNF) family of ATP-dependent chromatin remodeling proteins and the Structural Maintenance of Chromosomes (SMC) family, which we show associate with viral genomes in the absence of E4 proteins. We identified proteins from these families that exert a partial inhibitory impact on Ad5 infection by repressing viral genome replication and progeny production. Our study defines the repertoire of host factors that associate with Ad5 genomes and illustrates how these virus-host interactions are remodeled by early E4 proteins.

## RESULTS

**Whole-cell proteome analysis reveals dynamic alterations of host proteins and pathways during Ad5 infection.** To define changes to the cellular proteome during Ad5 infection, we compared uninfected (mock) human alveolar basal epithelial A549 cells to those infected with Ad5 WT over a time course of infection (Fig. 1). WCP samples were collected at 16, 24, and 48 h postinfection (hpi) and analyzed by label-free mass spectrometry (see Fig. S1A in the supplemental material). Approximately 6,000 to 7,000 proteins were quantified in each biological replicate, and protein abundances within the 4 replicates of each time point exhibited Pearson correlation coefficients of at least 0.96, indicating they were highly reproducible (Fig. S1B). Two-dimensional principal-component analysis (PCA) showed that the proteomes from each biological replicate clustered by infection time point (Fig. 2A). Differential expression analysis was performed to quantitate changes in protein abundance over the infection time course (see Materials and Methods for relevant fold change and significance thresholds). Relative to mock-infected cells, we observed that host proteins exhibited decreased abundance as Ad5 WT infection proceeded; 214 proteins decreased at 16 hpi, 659 at 24 hpi, and 2,103 at 48 hpi (Fig. 2B). Concurrently, we observed increased abundances for 533 proteins at 16 hpi, 433 at 24 hpi, and 887 at 48 hpi (Fig. 2B). Overall, 126 proteins decreased and 165 proteins increased in abundance at all time points (Fig. 2C). We also identified 45 viral proteins in the WCP analysis, most of which exhibited sustained or progressive increases over time (Fig. 2D and Fig. S1C). Detection of both early-expressed proteins (encoded by genomic regions E1, E2, E3, and E4) and late-expressed proteins (encoded by genomic regions L1, L2, L3, L4, and L5) confirmed appropriate progression through to the late stage of the viral infectious cycle in our experimental system. Altogether, these data demonstrate progressive changes to the host and viral proteomes throughout Ad5 infection.

As expected, we observed significant decreases in several host proteins known to be degradation substrates of the E4orf6-E1B55K virus-directed E3 ubiquitin ligase (Fig. 2E and F). These include DNA repair proteins MRE11, RAD50, and LIG4, the cellular receptor integrin a3, and the transcriptional regulator DAXX. The identification of LIG4 and DAXX under mock but not infection conditions prevented calculation of $\log_2$ fold change and associated $P$ values but supported that these proteins are degraded by Ad5 infection. Consistent with the observed decreases in our WCP data, several of the same proteins were previously found to be ubiquitinated upon ectopic expression of E1B55K and E4orf6 (31). This suggests that these decreased proteins may be targeted for degradation by the E4orf6-E1B55K E3 ligase complex. Additionally, we observed significant decreases in host proteins known to be targeted by E4orf3, including GTF2I, TRIM24, and TRIM33 (Fig. 2E and F).

The existence of known E4 targets decreased in the WCP data suggests that these data capture Ad5 manipulations of the host proteome and can be used to reveal additional E4-dependent impacts on cellular proteins and pathways. We therefore interrogated the WCP data specifically for proteins that are decreased during infection, and we hypothesized that these proteins are likely to include antiviral factors directly impacted by Ad5 infection. We observed distinct sets of proteins decreased at early (16 hpi), intermediate (24 hpi), and/or late (48 hpi) infection time points. The average abundance changes of the proteins within each group exhibit various dynamics, suggesting different mechanisms driving the abundance decreases and/or different biological relevance of the decreased proteins (Fig. 3A). To identify cellular processes that were manipulated at each of the 3 analyzed time points of Ad5 WT infection, Gene Ontology (GO) analysis was performed on the protein sets decreased in mock compared to 16 hpi, 24 hpi, or 48 hpi (Fig. 3A). All proteins significantly decreased at the respective time point were selected as inputs for the Gene Ontology (GO) analysis, including those that were also decreased at other time points. The resulting GO terms were filtered to include only those with statistically significant enrichment (false-discovery rate [FDR] < 0.05) and sorted by FDR values. The GO analysis of the top-10

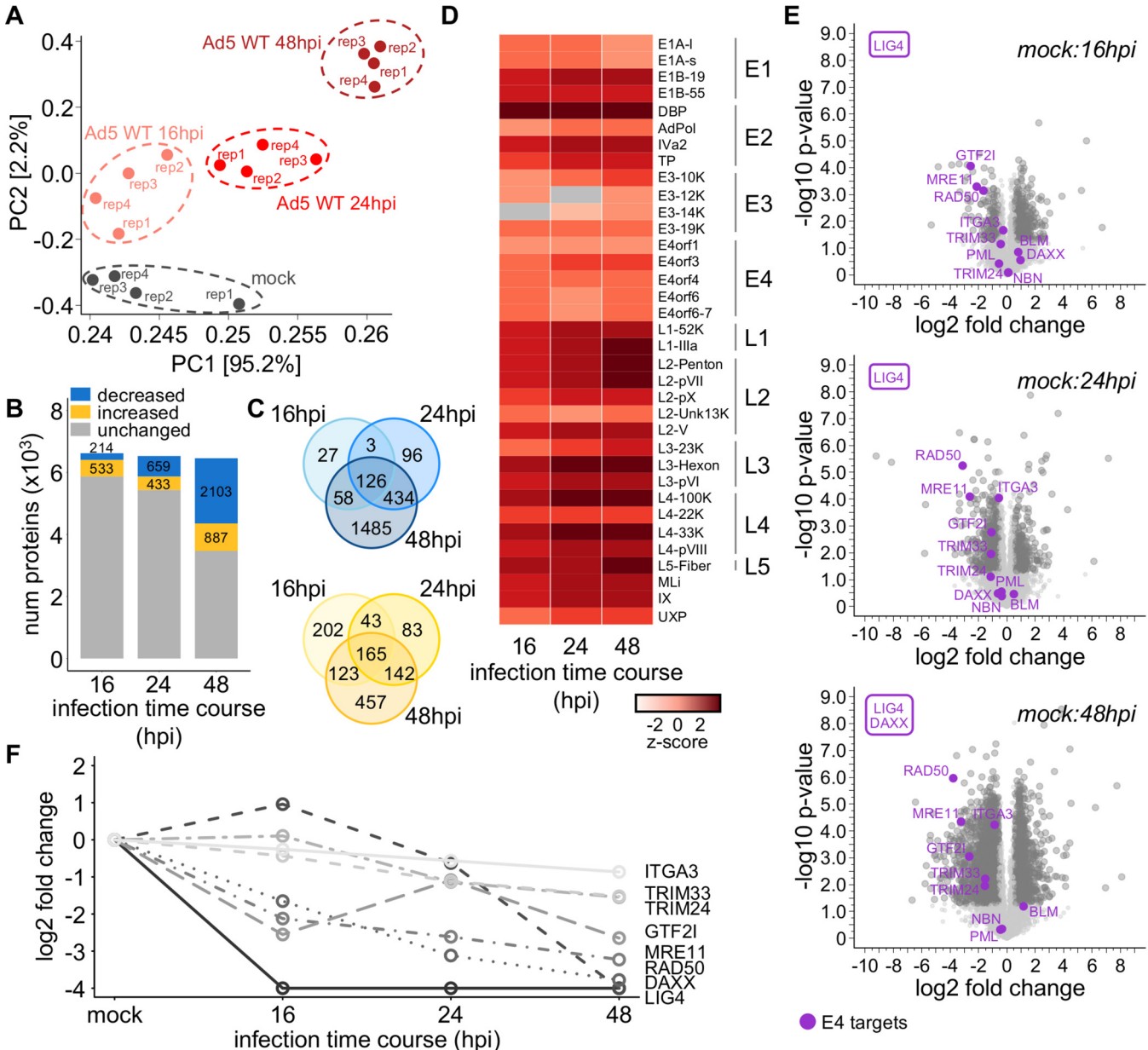

**FIG 2** WCP quantifies dynamic changes in host and viral proteomes during Ad5 WT infection. (A) Principal-component plot of the first two principal components generated from log$_2$-transformed normalized WCP quantification data of uninfected (mock) and Ad5 WT-infected samples at 16 hpi, 24 hpi, and 48 hpi. Clusters of biological replicates of similar infection time points are indicated. (B) Stacked bar plot showing numbers of proteins significantly decreased (blue), increased (gold), or unchanged (gray) at each time point of Ad5 WT infection, compared to uninfected mock samples. (C) Venn diagrams showing intersections of proteins decreased (upper Venn, blue colors) or increased (lower Venn, yellow colors), compared to mock, at each time point of Ad5 WT infection. (D) Heat map showing protein relative abundance z-score of canonical Ad5 WT viral proteins over the infection time course. Gray color indicates that the protein was not identified at that time point. Proteins are grouped according to the early and late transcription regions with which they are associated. (E) Volcano plots showing log$_2$ fold change and corresponding −log$_{10}$-transformed $P$ values for WCP-identified human proteins compared for abundance in 16-hpi (top), 24-hpi (middle), or 48-hpi (bottom) Ad5 WT infection time points compared to mock. Dark gray circles indicate proteins with statistically significant abundance differences between mock and the respective infection time point. Purple circles indicate known targets of E4orf6 or E4orf3. Proteins in purple frames (LIG4 and DAXX) are identified only under mock conditions and not identified in the respective time points, suggesting virus-induced decrease from the proteome. Since these proteins are quantified for mock but not the compared time point, there is no associated log$_2$ fold change or $P$ value corresponding to a point on the volcano plot. These proteins are included since they are known targets of Ad5 E4 proteins. (F) Line plot showing log$_2$ fold change in WCP abundance for proteins previously shown to be targeted by E4orf6 or E4orf3 viral proteins. All fold changes are normalized to mock, which is defined as 0-fold change. Log$_2$ fold change of −4 was imputed for proteins identified in mock but unidentified at one or more infection time points (LIG4 at 16 hpi, 24 hpi, and 48 hpi; DAXX at 48 hpi).

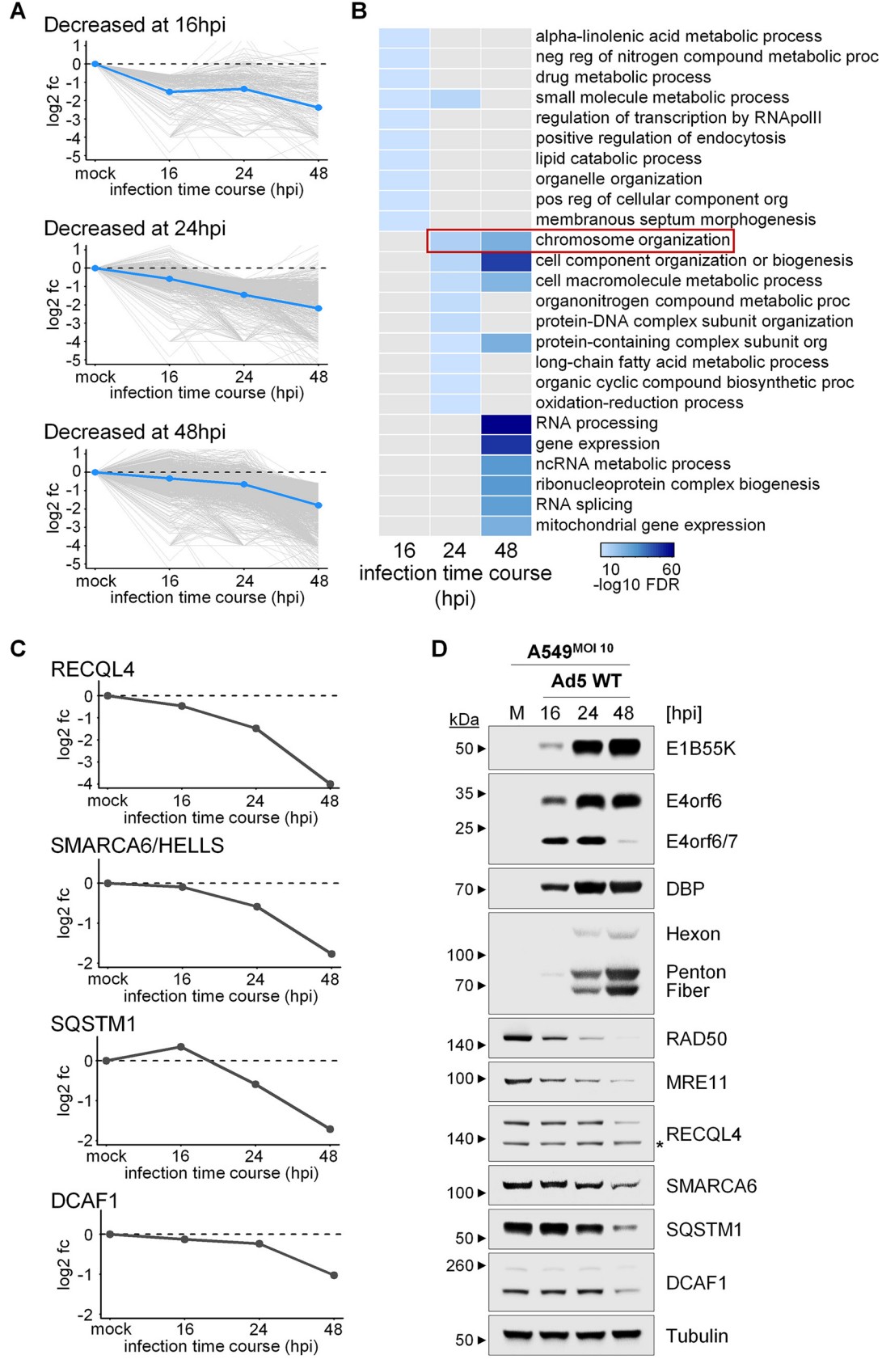

**FIG 3** Ad5 WT infection dynamically impacts cellular processes involving genome organization. (A) Line plots showing log₂ fold change in WCP abundance for proteins (gray lines) classified as significantly decreased at 16-hpi (top), 24-hpi (middle), or

most highly enriched terms within the decreased proteins at each time point suggests distinct cellular processes are impacted as infection progresses, including metabolism, chromosome organization, and RNA processing (Fig. 3B). These data suggest that Ad5 dynamically manipulates cellular processes at specific stages of viral infection.

We next validated the WCP results by immunoblotting. We probed for proteins from several enriched GO categories, including the "Chromosome organization" category since this process is the most highly enriched at 24 hpi and highly enriched at 48 hpi. Additionally, this category contains known AdV E4 targets MRE11, RAD50, and LIG4, suggesting that other decreased proteins in this category may be manipulated by AdV E4 proteins. We also reasoned that proteins from this category would be potentially relevant upon integration with iPOND data. Proteins in this category that exhibited decreases during infection included the SWI/SNF protein family member Lymphoid-specific helicase (HELLS/SMARCA6/LSH) and the ATP-dependent DNA helicase Q4 (RECQL4). The HELLS/SMARCA6/LSH protein cooperates with DNA methyltransferases and/or histone deacetylases to remodel nucleosomes, regulate DNA methylation, and repress transcription (46–48). RECQL4 is a member of the RecQ helicase family, which consists of BLM, WRN, RECQL1, RECQL4, and RECQL5. The RecQ family in general, including RECQL4, plays a role in genome maintenance in the DNA damage response (49). RECQL4 acts together with BLM to preserve genome stability (50). The RecQ family of proteins has been examined for potential manipulation by AdV, and it was shown that BLM is a degradation target of Ad5 E4orf6-E1B55K (27). While RECQL4 was not shown to be definitively targeted in the aforementioned study (27), we observed the most significant decreases at comparatively late time points in our WCP data set, and therefore, we further examined potential targeting of RECQL4. Additionally, we investigated the DCAF1 protein from the "RNA processing" category. This category was the most highly enriched cellular process at 48 hpi, and DCAF1 is an example of a protein that was decreased exclusively at the latest infection time point (48 hpi). We also included an assessment of the multifunctional scaffold protein Sequestosome-1 (SQSTM1/p62). Among the diverse roles for SQSTM1/p62 is its function as a receptor in selective autophagy (51). Autophagy plays an important role in the immune response to viruses, and autophagy-related proteins are antagonized by a variety of viruses (52). Indeed, SQSTM1/p62 has been shown to be degraded by the DNA virus herpes simplex virus 1 (HSV-1) viral E3 ligase ICP0 (53). The WCP data suggest that each of the selected proteins (HELLS/SMARCA6, RECQL4, SQSTM1, and DCAF1) are decreased during Ad5 WT infection, with distinct abundance change profiles (Fig. 3C). The immunoblot data also show that HELLS/SMARCA6, RECQL4, SQSTM1/p62, and DCAF1 each exhibit decreased protein abundances during Ad5 WT infection, consistent with the WCP data (Fig. 3D). These data provide experimental validation for changes in protein abundance during Ad5 WT infection identified in the WCP data set.

**FIG 3** Legend (Continued)

48-hpi (bottom) Ad5 WT infection compared to mock. Blue line shows average fold change of all proteins within the specific classification, at the respective time point. All fold changes are normalized to mock, which is defined as 0-fold change. Log$_2$ fold change of $-4$ was imputed for proteins identified in mock but unidentified at one or more infection time points. (B) Heatmap showing the 10 most enriched Gene Ontology (GO) Biological Processes terms identified within the WCP proteins that are significantly decreased at 16 hpi, 24 hpi, or 48 hpi compared to mock. GO analysis was performed using the functional annotation calculated from the STRING database in the Cytoscape network analysis software. GO terms were filtered for 50% redundancy using the STRING functional annotation analysis. Resulting terms were selected to include only those exhibiting statistically significant enrichment (FDR $<$ 0.05) and sorted by smallest FDR. Heatmap colors correspond to $-\log_{10}$-transformed FDR values. GO terms with colors at multiple time points indicate that the term was in the top-10 enriched GO Biological Processes for the proteins decreased at each time point. Gray colors indicate that the term was not in the top-10 enriched GO Biological Processes for the proteins decreased at that time point. Some terms are abbreviated for figure clarity. (C) Line plot showing log$_2$ fold change in WCP abundance for proteins selected for experimental interrogation of protein abundance by immunoblot analysis. All fold changes are normalized to mock, which is defined as 0-fold change. (D) Immunoblot analysis of A549 cell infection with Ad5 WT at an MOI of 10. Viral early proteins (E1B55K, E4orf6, E4orf6/7, DBP) and late proteins (hexon, penton, fiber) increase as expected over the time course of infection. Known degraded substrates of the E4orf6-E1B55K complex, RAD50 and MRE11, are decreased during Ad5 WT infection. Cells were harvested at the indicated times postinfection. Proteins were detected with antibodies specific to viral proteins (E1B55K, E4orf6, DBP, and late proteins hexon, penton, and fiber) or the indicated cellular proteins. Tubulin served as a loading control. Molecular weight markers are indicated to the left. Asterisk, nonspecific band.

We next assessed whether the decreases in protein abundance we observed for HELLS/SMARCA6, RECQL4, SQSTM1/p62, and DCAF1 are dependent on the E4 genomic region of Ad5. To determine the dependence on the E4 genomic region, we used immunoblotting to compare protein abundances for selected proteins during a time course of infection with Ad5 WT and AdΔE4 mutant virus. The decreases in abundance observed for HELLS/SMARCA6, RECQL4, SQSTM1/p62, and DCAF1 during Ad5 WT infection were mitigated during AdΔE4 infection, suggesting that manipulation of these proteins is dependent on one or more E4 gene products (Fig. 4A). Since the E4orf6-E1B55K complex induces degradation of host antiviral factors, we assessed the role of this viral E3 ligase complex in degrading selected proteins. Immunoblot analysis in infected cells treated with a neddylation inhibitor (54), which inhibits cullin-mediated ubiquitination, was performed to determine whether the observed decreases in abundance are dependent on ubiquitination by a cullin complex. Neddylation inhibition induces a shift of the CUL5 protein band to a lower molecular weight, reflecting loss of neddylation (Fig. 4B). The neddylation inhibitor blocks the activity of the virus-directed ligase and therefore rescues degradation of known E4orf6-E1B55K substrates RAD50 and MRE11 (Fig. 4B). Additionally, neddylation inhibition induces a partial mitigation of decrease in protein abundance of HELLS/SMARCA6 and SQSTM1/p62 during Ad5 WT infection (Fig. 4B). This suggests that the decrease in these two proteins is the result of cullin-based ubiquitination and degradation. To investigate the possibility that the decreased proteins HELLS/SMARCA6, RECQL1, SQSTM1/p62, and DCAF1 are degraded in an E4orf6-E1B55K-dependent manner, we infected cells with either AdV ΔE4orf6 or ΔE1B55K mutant viruses. Each mutant virus infection abrogates degradation of known substrates RAD50 and MRE11 (Fig. 4C and D and Fig. S2). The ΔE4orf6 or ΔE1B55K viruses each exhibit partial rescue of HELLS/SMARCA6, RECQL4, SQSTM1/p62, and DCAF1 levels (Fig. 4C and D and Fig. S2). Together, these data suggest that proteins identified as decreased in the WCP include degraded substrates of the E4orf6-E1B55K complex. However, the fact that we observe a nearly complete rescue of protein levels during infection with the AdΔE4 virus but only a partial rescue of the protein levels during infection with the ΔE4orf6 or ΔE1B55K mutant viruses suggests that there may be some redundancy among E4 proteins in targeting these cellular factors. The E4orf3 protein is a likely contributor.

In summary, our WCP data quantitate the impact of Ad5 infection on cellular proteins and pathways. We identify host proteins that are decreased in abundance, some of which are dependent on the E4 viral genomic region. The decreases observed in the WCP data include known targets of E4 proteins, as well as previously unknown targets of E4.

**Host proteins differentially associate with replicating Ad5 genomes in the absence of E4.** Host antiviral proteins and restriction factors can target viral genomes to inhibit viral processes. Moreover, we observe in the WCP data that cellular processes such as "Chromosome organization" and "Protein-DNA complexes" are manipulated in an E4-dependent manner. Therefore, we devised an iPOND-based proteomics strategy to further define the repertoire of host proteins recruited to AdV genomes during infection with WT virus or in the absence of E4 gene products with the AdΔE4 mutant virus (Fig. 1). We hypothesized that host proteins enriched on AdΔE4 viral genomes include cellular proteins that likely act in an antiviral capacity and are counteracted by one or more of the E4 proteins during Ad5 WT infection. We expected these host proteins to be degraded or mislocalized by E4 proteins and would therefore not associate with WT genomes. iPOND is based on labeling replicating DNA with a deoxythymidine analog, in this case 5-ethynyl-2-deoxyuridine (EdU). The labeled DNA is biotinylated by click chemistry, and streptavidin bead purification is used to isolate the DNA and formaldehyde-cross-linked proteins (43, 55). Mass spectrometry is used to identify and quantify the isolated proteins (41). iPOND was performed on uninfected U2OS cells (mock) or cells infected with Ad5 WT or AdΔE4 mutant virus at 24 hpi and analyzed by mass spectrometry in data-dependent acquisition (DDA) mode (Fig. S3A). The iPOND proteomics technique quantified approximately 2,000 to 2,500 DNA-associated proteins in each biological replicate, with a total of 2,884 proteins identified in the entire experiment (Fig. S3B). The protein abundance data within the 3 replicates of each

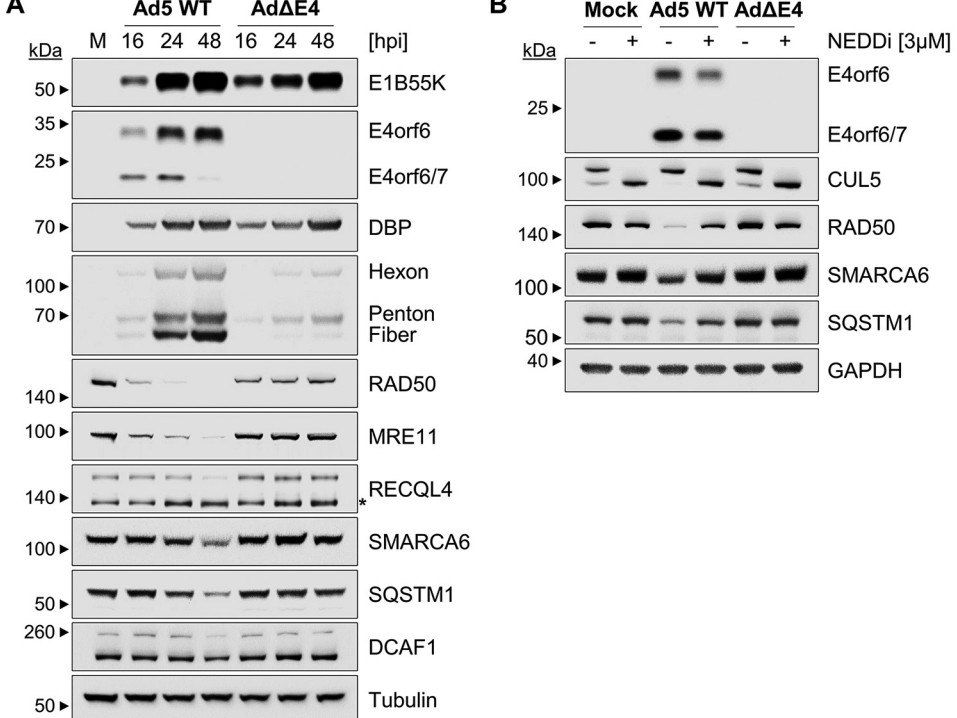

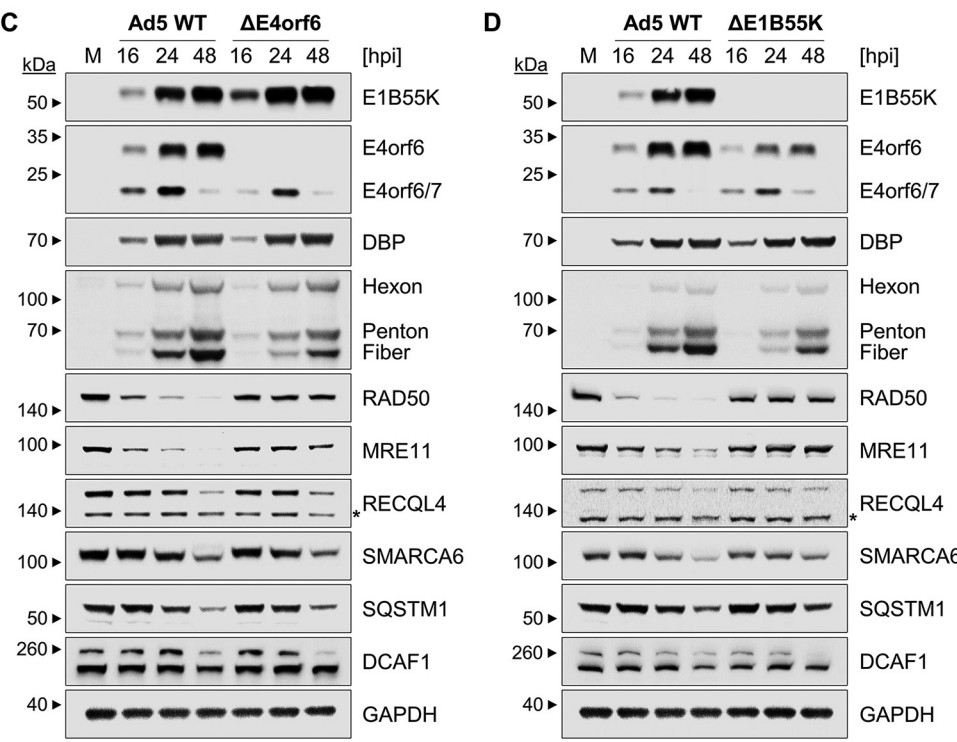

**FIG 4** Contributions of neddylation and the viral E4orf6-E1B55K complex to proteome changes. (A to D) Immunoblot analysis of A549 cells infected at an MOI of 10. (A) Cells infected with Ad5 WT compared to either AdΔE4 (A and B), AdΔE4orf6 (C), or AdΔE1B55K (D) and harvested at the indicated times postinfection. (B) Cells were harvested at 24 hpi in the presence of DMSO or neddylation inhibitor MLN4924 (3 μM), added at 8 hpi. Viral early proteins (E1B55K, E4orf6, E4orf6/7, DBP) and late proteins (hexon, penton, fiber) increase as expected over the time course of Ad5 WT infection. Known degraded substrates of the E4orf6-E1B55K complex, RAD50 and MRE11, are decreased during Ad5 WT infection. Infection with mutant viruses impacts protein abundance of relevant viral proteins and known degraded substrates of the E4orf6-E1B55K complex. Tubulin and GAPDH served as loading controls. Molecular weight markers are indicated to the left. Asterisk, nonspecific band.

infection condition are highly reproducible, exhibiting Pearson correlation coefficients of at least 0.85 (Fig. S3B). Principal-component analysis (PCA) shows that the proteomes generated from each biological replicate cluster according to their respective infection condition (Fig. 5A). Furthermore, the clusters of biological replicates occupy distinct regions of the two-dimensional PCA plot, suggesting that isolated proteomes for each condition are globally disparate and likely contain proteins that are differentially enriched on AdΔE4 or Ad5 WT viral genomes (Fig. 5A).

We hypothesized that Ad5 viral gene products, such as those of the E4 genomic region, act to impact host proteins associated with viral genomes. Since E4 viral proteins can induce degradation or mislocalization of host proteins such as the MRN complex (25, 35) and PML (32, 33), we hypothesized that antiviral proteins may be recruited to viral genomes during AdΔE4 infection. Conversely, E4 proteins may impact recruitment of proviral host factors onto viral genomes to promote viral processes. Thus, we expected that host proteins enriched on viral genomes during AdΔE4 or WT infection would reflect a combination of targeting and recruitment by E4 viral proteins. Differential expression analysis was performed to classify proteins as enriched on either of the Ad5 WT or AdΔE4 genomes based on comparative fold change, or by being identified during Ad5 WT or AdΔE4 infection exclusively (see Materials and Methods for relevant fold change and significance thresholds). Quantitative comparison of proteins associated on the Ad5 WT or AdΔE4 genomes reveals that 264 proteins are enriched on AdΔE4 genomes compared to Ad5 WT, while 236 proteins are enriched on the Ad5 WT genome (Fig. 5B). These data suggest that distinct subsets of cellular proteins associate with Ad5 WT or AdΔE4 viral genomes.

The Ad5 E4orf6-E1B55K E3 ubiquitin ligase (Fig. S3C) targets cellular antiviral proteins for proteasome-mediated degradation, including the MRE11, RAD50, and NBN proteins that form the MRN complex (24, 25, 56–59). The MRN complex functions in cellular double-strand break repair and homologous recombination and also recognizes replicating viral DNA (60). Direct comparison of the proteomes associated with Ad5 WT and AdΔE4 genomes shows that each of the MRN complex proteins, GTF2I, TRIM33, and PML, are differentially enriched on AdΔE4 genomes compared to WT (Fig. 5C and Fig. S3D). The relative abundance of these proteins within the iPOND proteome of AdΔE4 infection is similar to uninfected cells, suggesting that these proteins are associated with nascent DNA of host cell and AdΔE4 genomes at similar levels (Fig. S3D). Furthermore, the MRN complex proteins, as well as GTF2I, TRIM33, and PML, exhibit high relative abundance on AdΔE4 genomes compared to WT genomes (Fig. 5D). These data suggest that antiviral proteins antagonized by E4 viral gene products are enriched on AdΔE4 genomes. The characteristic abundance patterns observed for known antiviral proteins could be used to inform predictions of novel antiviral restriction factors of Ad5 from the iPOND data.

GO analysis shows disparate cellular pathways represented for the top-10 most highly enriched processes within the proteomes associated with Ad5 WT or AdΔE4 genomes (Fig. 5E and Fig. S3E). These cellular processes and pathways enriched within the proteins differentially associated with the AdΔE4 genomes may exert antiviral effects during Ad5 infection and may be counteracted by E4 viral gene products during WT infection. The GO processes enriched within the proteins associated with AdΔE4 include the most highly enriched "DNA metabolic process," which contains terms such as "DNA double-strand break processing" and "DNA repair" within its hierarchy. These processes are related to the DNA damage response pathway, a known component of the host antiviral response, which is counteracted by E4 gene products during Ad5 infection (61). Interestingly, the second most highly enriched GO process is "Chromosome organization," which describes processes related to arrangement of DNA and associated proteins, covalent modifications, and maintenance of genome integrity. The iPOND GO analysis and WCP results together suggest that cellular processes involving "Chromosome organization" are likely manipulated during Ad5 infection via E4 gene products targeting relevant host proteins. In order to identify proteins that are antagonized by E4 gene products to

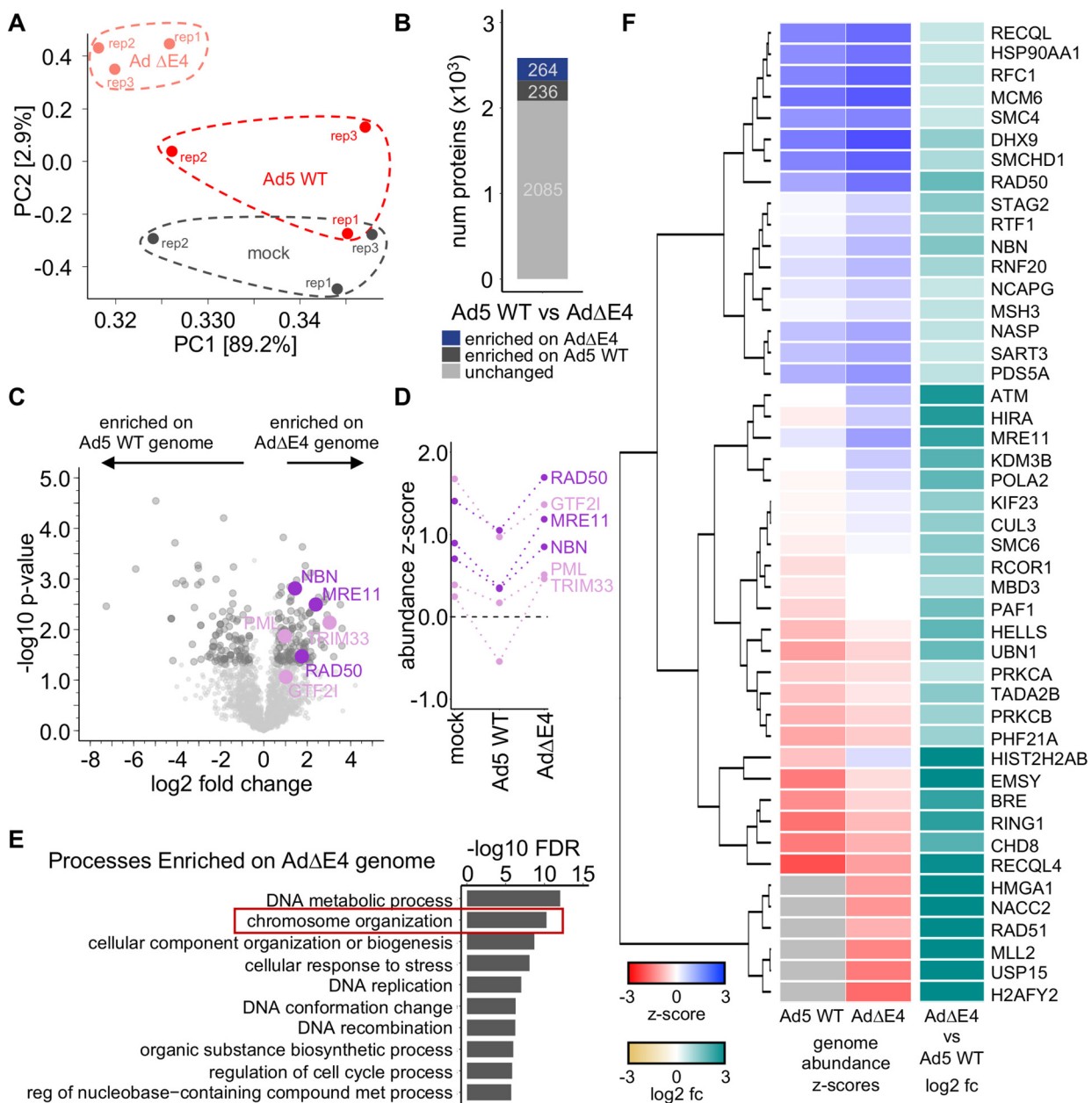

**FIG 5** iPOND identifies proteins associated with Ad5 genomes in the absence of E4 gene products. (A) Principal-component plot of the first two principal components generated from the mock, Ad5 WT, and AdΔE4 log$_2$-transformed, normalized iPOND proteome quantification data, for all proteins identified with at least 1 unique peptide, from infected U2OS cells (MOI of 40). Clusters of biological replicates of similar infections are indicated. (B) Stacked bar plot showing numbers of proteins enriched on the AdΔE4 genome (blue), enriched on the Ad5 WT genome (dark gray), or unchanged (light gray) for comparison of Ad5 WT and AdΔE4 iPOND abundance data. (C) Volcano plot showing log$_2$ fold change and corresponding $-\log_{10}$-transformed $P$ values for iPOND-identified proteins compared for abundance on AdΔE4 genomes compared to Ad5 WT. Dark gray circles indicate proteins with statistically significant abundance differences between Ad5 WT and AdΔE4 genomes. Proteins previously shown to be targets of E4orf6 (dark purple) or E4orf3 (light purple) are indicated and exhibit increased abundance on the AdΔE4 genome. All differences are statistically significant ($P < 0.05$) except for GTF2I ($P < 0.08$). (D) Relative abundance on mock, Ad5 WT, and AdΔE4 genomes, as measured by statistical z-score, for known targets of E4orf6 (dark purple) and E4orf3 (light purple). Together, panels C and D show a characteristic increase in abundance on the AdΔE4 compared to Ad5 WT genome, and a high relative abundance on the AdΔE4 genome, for proteins antagonized by E4orf6 or E4orf3. (E) Bar graph showing the 10 most enriched GO Biological Processes identified within the iPOND proteins that are enriched on the AdΔE4 genome compared to Ad5 WT. GO analysis was performed using the functional annotation calculated from the STRING database in the Cytoscape network analysis software. GO terms were filtered for 50% redundancy using the STRING functional annotation analysis. Resulting terms were selected to include only those exhibiting statistically significant enrichment (FDR < 0.05) and sorted by smallest FDR. Bar lengths correspond to $-\log_{10}$-transformed FDR values. Some terms are abbreviated for figure clarity. (F) Heatmap showing the iPOND protein relative genome abundance z-scores on the Ad5 WT and AdΔE4 genomes (left and center columns), as well as the log$_2$ fold change of iPOND abundance for AdΔE4 compared to Ad5 WT (right column), for the 46 proteins enriched on the AdΔE4 and annotated to constitute the "Chromosome Organization" GO Biological Process. Gray color indicates protein was not identified for that experiment.

prevent restriction of virus infection, we compared the iPOND genome abundances (z-score) with the relative fold change for the "Chromosome organization" proteins enriched on AdΔE4 genomes (Fig. 5F). The protein HELLS/SMARCA6, which is decreased in the WCP data, is also found enriched on AdΔE4 genomes. The SMC family proteins SMC4 and SMCHD1 are clustered with RAD50, a known target of the E4orf6-E1B55K viral E3 ligase (25). Additionally, a third member of the SMC family, SMC6, is also enriched on AdΔE4 genomes. We hypothesized that these proteins may be important for inhibiting viral infection by associating with the genome and are therefore counteracted by E4 viral proteins. Alternatively, proteins could be enriched on the AdΔE4 due to recognition of defects in viral genome integrity due to replication deficiencies and/or associated DNA damage responses.

In summary, quantitating the host proteomes associated with Ad5 WT and AdΔE4 viral genomes suggests that a subset of cellular proteins associate with viral genomes in the absence of E4 countermeasures. The presence of multiple members of the SWI/SNF and SMC protein families within the proteins differentially enriched on AdΔE4 genomes suggests that one or more proteins within these families may target viral genomes to restrict infection and are counteracted by E4 viral gene products during Ad5 infection.

**SWI/SNF and SMC proteins impact Ad5 infection.** We integrated the WCP and iPOND data for all the identified SWI/SNF and SMC family members to identify specific proteins that may be localized to Ad5 genomes and/or degraded from the cellular proteome in an E4-dependent manner (Fig. 6A). The HELLS/SMARCA6 protein is shown by WCP and immunoblotting to be decreased in an E4-dependent manner. The WCP data suggest that SMARCAD1 is not significantly decreased at 24 hpi. These data suggest that HELLS/SMARCA6 and SMARCAD1 are both enriched on AdΔE4 genomes, based on $\log_2$ fold change in iPOND abundance. Although SMARCAD1 exhibits a relatively high $\log_2$ fold change in iPOND abundance increase on AdΔE4 genomes compared to Ad5 WT genomes, it is not classified as enriched on AdΔE4 viral genomes due to lack of statistically significant hypothesis test results ($P < 0.16$). To explore further the potential association of HELLS/SMARCA6 and/or SMARCAD1 with viral genomes, we assessed the intracellular localization of these proteins during mock, Ad5 WT, and AdΔE4 infections using immunofluorescence (IF) microscopy. The location of VRCs were marked by immunostaining of the viral DNA-binding protein (DBP) and of bromodeoxyuridine (BrdU) incorporated into newly replicated viral DNA by pulse-labeling (Fig. 6B; see also Fig. S4A and Fig. S5). HELLS/SMARCA6 exhibits diffuse nuclear localization without obvious colocalization with DBP during infection with Ad5 WT or AdΔE4 viruses (data not shown). SMARCAD1 exhibits diffuse nuclear localization in uninfected cells and remains broadly nucleoplasmic, during Ad5 WT infection. In contrast, during AdΔE4 infection, the SMARCAD1 protein exhibits striking reorganization to colocalize with DBP and BrdU at VRCs (Fig. 6B; see also Fig. S4A and Fig. S5). These data suggest that SMARCAD1 association with viral genomes is limited in the presence of E4 proteins. Although SMC5 and SMC6, which form a complex, are associated with AdΔE4 viral genomes in the iPOND data (Fig. 6A), only SMC6 reaches statistical significance. We assessed the localization of SMC6 during mock, Ad5 WT, and AdΔE4 infections using IF. SMC6 exhibits diffuse nuclear and occasional perinuclear localizations during mock or Ad5 WT infection but exhibits striking reorganization to colocalize with DBP and BrdU at VRCs during AdΔE4 infection (Fig. 6B; see also Fig. S4A and Fig. S5). Consistent with the statistical results, SMC5 did not significantly relocalize during AdΔE4 infection relative to mock or Ad5 WT infections (Fig. S4B).

The iPOND and IF data suggest that SMARCAD1 and SMC6 localize to viral genomes, at VRCs, during infection with AdΔE4 but not with Ad5 WT virus. The fact that these proteins localize to VRCs and are associated with viral genomes in the absence of E4 gene products, but not during WT infection, suggests that they are manipulated by E4 viral proteins to counteract an inhibitory host response. To test the potential inhibitory impact of SMARCAD1 and SMC6 proteins on Ad5 infection, we used small interfering RNA (siRNA)

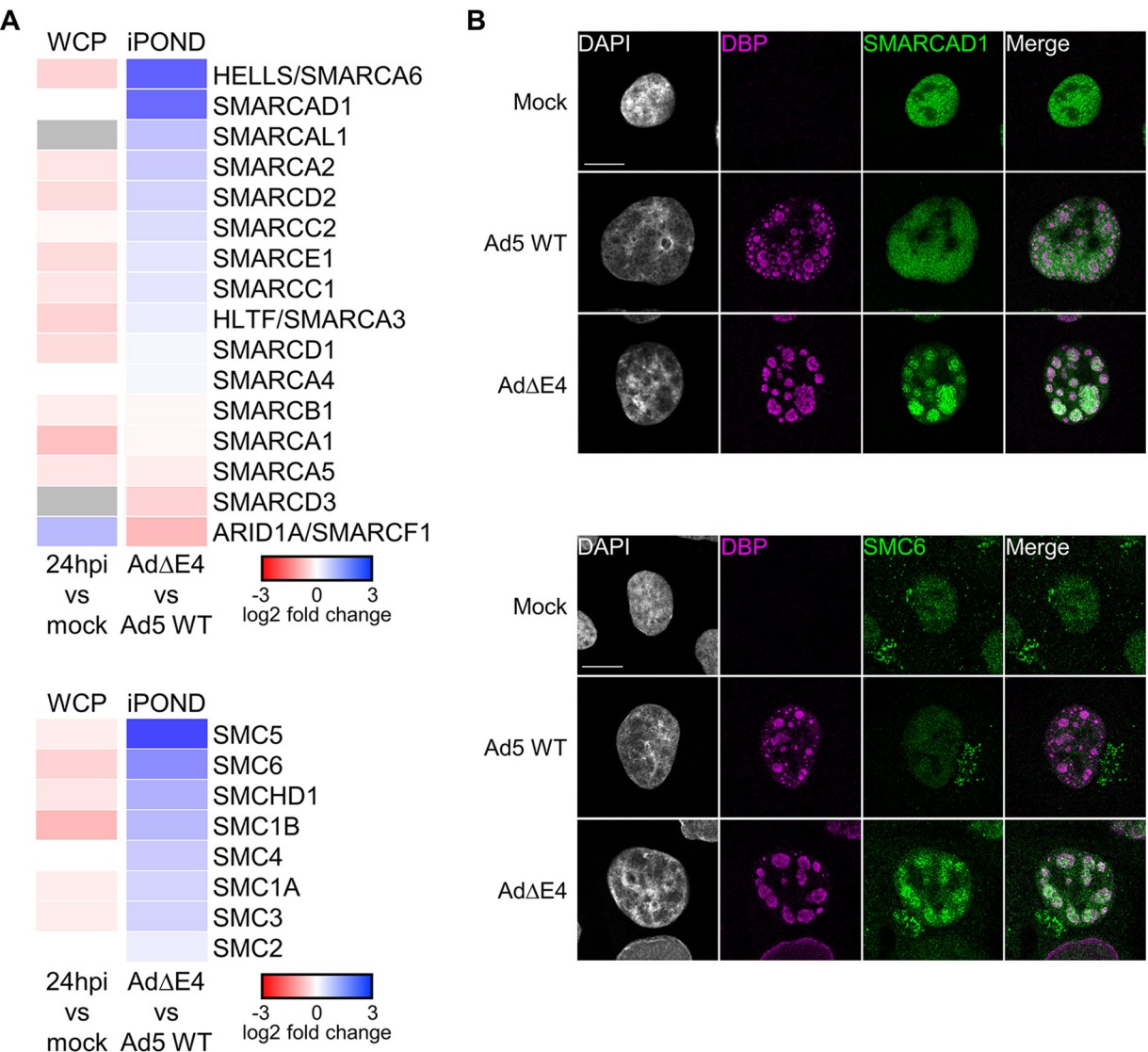

**FIG 6** Integration of WCP and iPOND data sets defines E4-dependent manipulation of chromosome organization proteins by Ad5. (A) Heatmap showing the $\log_2$ fold change in WCP abundance at 24 hpi compared to mock and the $\log_2$ fold change of iPOND abundance for AdΔE4 compared to Ad5 WT, for each of the iPOND-identified proteins within the SWI/SNF (top) or SMC (bottom) protein families. Gray color indicates protein was not identified for that experiment. (B) Immunofluorescence microscopy showing SMARCAD1 (top) or SMC6 (bottom) accumulation at nuclear viral replication centers in the absence of Ad5 E4 gene products. For IF experiments shown in panel B, cellular proteins SMARCAD1 and SMC6 are shown in green and viral DNA binding protein DBP is shown in magenta. A549 cells were either uninfected or infected with Ad5 WT or AdΔE4 at an MOI of 40 and analyzed at 24 hpi. The viral DBP indicates sites of viral replication centers in the nucleus. DAPI marks host cell nuclei. Scale bar = 10 μm.

to deplete them individually prior to infection. Since SMC5 is known to form a complex with SMC6, we also tested the impact of SMC5 depletion by siRNA on infection. We then measured fold change in viral genome accumulation during Ad5 WT and AdΔE4 infections (4, 24, and 48 hpi). Depletion of SMARCAD1 or SMC5 is associated with statistically significant increases in the number of viral genomes at 48 hpi in Ad5 WT infection compared to control siRNA treatment (Fig. 7A). Depletion of SMC6, however, is associated with a slight change in number of viral genomes during Ad5 WT infection, compared to control siRNA treatment (Fig. 7A). During AdΔE4 infection, depletion of SMARCAD1 is associated with no change in number of viral genomes, while depletion of either SMC5 or SMC6 is associated with statistically significant increases in viral genomes at 48 hpi compared to control siRNA treatment (Fig. 7A). Since SMARCAD1 is differentially enriched on viral genomes in the absence of E4 gene products and appears to show inhibitory

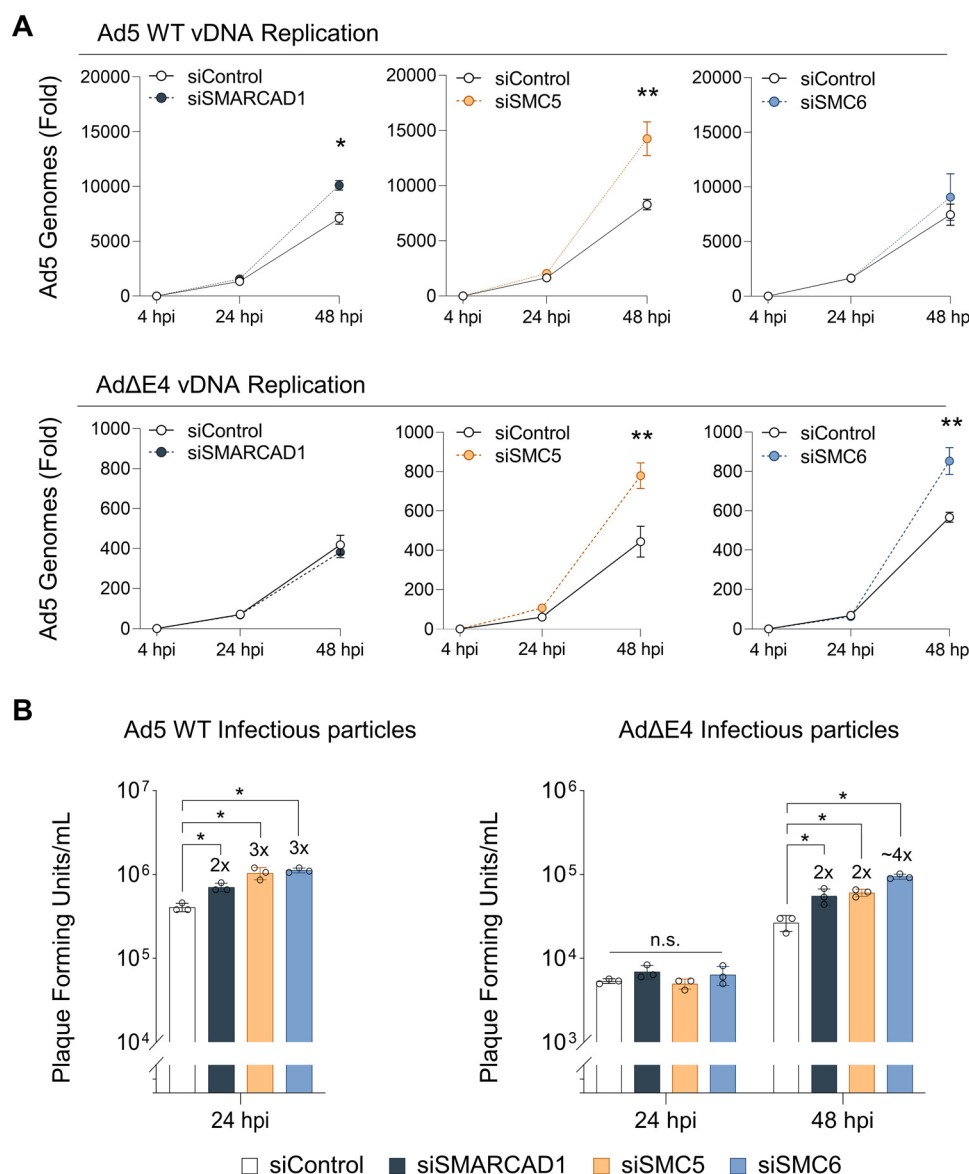

**FIG 7** SMARCAD1, SMC5, and SMC6 impact viral DNA replication and progeny production. A549 cells depleted of SMARCAD1, SMC5, or SMC6 by siRNA were infected with Ad5 WT or AdΔE4 (MOI of 10) over a time course of 4 hpi, 24 hpi, and 48 hpi. (A) Viral DNA replication was measured by qPCR in biological triplicate at the indicated times postinfection (hpi). Genome copies at 24 hpi or 48 hpi were quantified by fold change normalization to input DNA at 4 hpi in cells treated with control siRNA. (B) Infectious viral particle production from infected A549 cells was assessed in W162 cells at 24 hpi and/or 48 hpi by plaque assay. For all assays, statistical significance was determined via unpaired two-tailed Student's $t$ test (*, $P \leq 0.05$; **, $P \leq 0.01$; n.s., not significant). Experiments represent three biological replicates with graphs depicting the mean (error bars, ±standard deviation).

impacts in WT infection, we expected that depletion in the context of AdΔE4 infection would also result in increased number of viral genomes. The fact that we did not observe a significant increase in viral genomes for this condition may be explained by functional redundancy within the SWI/SNF protein family or other antiviral proteins. Depletion of SMARCAD1 may not have been sufficient to overcome the deficiencies in counteracting host defenses that occur during AdΔE4 infection. We next assessed whether the increase in viral genome number during siRNA-mediated depletion of SMARCAD1, SMC5, or SMC6 was manifested in an increase in production of viral progeny, as measured by plaque assay. During Ad5 WT infection, depletion of SMARCAD1, SMC5, or SMC6 is associated with 2- to 3-fold, statistically significant increases in PFU at 24 hpi compared to control

siRNA treatment (Fig. 7B). During AdΔE4 infection, depletion of SMARCAD1, SMC5, or SMC6 is associated with no change in PFU at 24 hpi but is associated with 2- to 4-fold, statistically significant increases in PFU at 48 hpi compared to control siRNA treatment (Fig. 7B). We used immunoblotting to test the impact of depletion of SMARCAD1, SMC5, or SMC6 on expression of late viral proteins. During Ad5 WT or AdΔE4 infection, Ad5 hexon, penton, and fiber proteins show no meaningful change in abundance in the context of depletion of SMARCAD1, SMC5, or SMC6 compared to control siRNA treatment (Fig. S6A and B). The immunoblotting data suggest that SMARCAD1 and SMC6 localizations at AdΔE4 viral replication centers are not driven by increased abundance during AdΔE4 infection. The data assessing the impact of depletion of SMARCAD1, SMC5, or SMC6 on Ad5 infection suggest that these proteins exert inhibitory effects on productive Ad5 infection, which are more pronounced during AdΔE4 infection. Although the observed impacts of individual depletion of SMARCAD1, SMC5, or SMC6 were modest, this could be explained by redundancy within the host response since each of these proteins is part of families in which multiple members may be acting in a coordinated manner to inhibit infection.

## DISCUSSION

In this study, we integrated quantitative mass spectrometry approaches to profile cellular factors and processes manipulated by Ad5 infection. We analyzed alterations in the host proteome and associations with the viral genome during infection. Our WCP data measure protein abundance changes in response to Ad WT infection and suggest subsets of proteins that are dynamically manipulated over a time course of infection. Our iPOND comparison between Ad5 WT and AdΔE4 genomes provides a catalog of viral genome-associated proteomes that are remodeled in the absence of E4 gene products. The differences observed in the iPOND proteome data reflect deficiencies in the AdΔE4 mutant virus, such as an inability to counteract host defenses or redirect cellular processes necessary for productive infection. We demonstrate the utility of our approach by integrating these proteomics data sets and performing a systems-level analysis. We identified that (i) RECQL4, SMARCA6, SQSTM1, and DCAF1 are degradation targets of Ad5 early proteins; (ii) SMARCAD1 and SMC6 are associated with replicating DNA genomes in the absence of Ad5 E4 early proteins; and (iii) SMARCAD1 and SMC family components SMC5 and SMC6 exert subtle inhibitory effects on infection. Our findings suggest that Ad5 employs multiple strategies to antagonize cellular factors which have shared functions in chromatin remodeling and maintaining genome stability.

Previous studies have employed proteomics to define interactions and targets of early AdV proteins (31, 62–67). However, most studies that have explored degradation targets focused on early stages of infection. Our WCP study extends knowledge of alterations within the host proteome into the late phase of infection. The greater depth of our WCP data enabled us to reproducibly capture a broad range of proteome abundance changes during infection, including expected increases for viral proteins and decreases for cellular degradation targets. Upon follow-up experimentation by immunoblotting, we reproducibly validated both large and subtle abundance decreases. In doing so, we identified a temporally distinct subset of cellular degradation targets which correlate with the late phase of infection. For example, we observed that RAD50 abundance decreased 8.7-fold relative to mock-infected cells by 24 hpi and 13.7-fold by 48 hpi. Distinct from this, DCAF1 minimally decreased 1.2-fold by 24 hpi, yet decreased 2-fold with statistical significance by 48 hpi. Despite these differences, each of these proteins decreased in an E4-dependent manner. Infection with AdΔE4 allowed us to implicate a significant role for the E4 cassette in the degradation of the host factors DCAF1, RECQL4, SMARCA6, and SQSTM1. We also observed only partial defects in degradation upon infection with mutant Ad5 strains lacking components of the virus-formed E4orf6-E1B55K E3 ubiquitin ligase complex. Collectively, these observations suggest that one or more E4 gene products coordinate with other viral or cellular factors to facilitate efficient degradation. It may also be that coordination is engaged

primarily during the late phase of infection. Consistent with this, the degradation kinetics of the novel targets are similar to those of previously established Ad5 E4orf3-driven degradation targets (GTF2I, TRIM33, TRIM24) (37, 38). Thus, it is possible that E4orf3 or another E4 gene product contributes to the observed degradation. The temporal regulation of the host proteome may reflect the dependency on additional post-translational modifications. Alternatively, late viral protein expression and progression through the late phase of infection may be predominant factors dictating the observed E4 dependency. Furthermore, host transcriptional responses may compensate for virus-induced degradation of some cellular proteins, resulting in less substantial abundance decreases observed at later times of infection. Future systematic investigations are warranted to define the contribution of each E4 protein in promoting efficient degradation of the uncovered cellular degradation targets, including analysis of E4 protein-protein interactions throughout infection.

Host proteins exhibiting enriched association with AdΔE4 genomes may be recruited as a consequence of DNA damage responses and signaling induced by mutant viruses (24, 61, 68, 69). Our iPOND data did indeed show that the MRN complex and ATM kinase were highly enriched on AdΔE4 genomes relative to Ad5 WT. Along with these known DDR factors, which have been previously demonstrated to interfere with AdV replication, we identified multiple cellular components involved in the DDR and chromatin regulation enriched on AdΔE4 genomes. Among these were proteins from the SWI/SNF-related (SMARCAD1) and the SMC (SMC6) families, each known to have roles in maintaining chromosome integrity (70–77). The various degrees of association that we observed may indicate temporal regulation and/or functional redundancy. SMARCAD1 has been found to function in both homologous recombination (HR)-mediated DNA double-strand break (DSB) repair and the reestablishment of a heterochromatin state by promoting histone methylation (71, 78). It was suggested that SMARCAD1 displaces a chromatin barrier, 53BP1, from DNA and recruits HR components to facilitate DNA end-resection. In another study, recruitment of SMARCAD1 to sites of DSBs was suggested to be dependent on the activity of a core protein involved in DNA damage responses, ataxia-telangiectasia mutated (ATM) (72). Since ATM activity is known to interfere with DNA replication of Ad strains deficient in E4 components (79), SMARCAD1 may contribute to viral restriction downstream of ATM. We note that another SWI/SNF-related protein, SMARCAL1, was observed to be both targeted for degradation by Ad5 via an interaction with E1B55K and localized to VRCs (80). Recruitment to VRCs was suggested to occur primarily via the RPA complex, a core macromolecular signaling hub involved in DNA replication and repair. As such, SMARCA6 and SMARCAL1 may be targeted for degradation, while SMARCAD1 may be constrained from viral genome association through redundant means to modulate the progression of viral genome replication and progeny production.

Reports in the literature also provide evidence for virus manipulation of SMC5 and SMC6 (81–89). As part of a complex, SMC5/6 was suggested to silence expression of unintegrated HIV-1 DNA by promoting the loss of active histone methylation marks (83). During infection with hepatitis B virus (HBV), SMC5/6 was shown to suppress viral gene transcription by associating with the viral DNA genome (87). This antiviral activity was further shown to be antagonized by the HBV X protein, which was observed to target SMC6 for degradation (87, 88, 90). During infection with human papillomavirus, SMC5/6 was found to localize to viral replication foci and depletion of SMC6 decreased total viral DNA levels (82). These studies raise the possibility that SMC5 and SMC6 may engage in different activities with viral DNA, which is consistent with our findings. The effects of the SMC5/6 complex may be the result of repressive chromatin or direct inhibition of viral DNA replication. As functions of the SMC5/6 complex are still emerging (91), comparative infections of cells with Ad5, HIV, HBV, and papillomavirus may be leveraged as virus models to explore the endogenous roles of SMC5/6. To begin assessing for functional redundancy, it will be critical to determine the impact of depleting multiple components of each protein group (SWI/SNF-related or SMC) on Ad5 infection.

The results of this study provide novel examples of host factors that are targeted for degradation in an E4-dependent manner and associate with the viral genome in the absence of the E4 gene products. We anticipate that the described data will also serve as a resource for additional studies of Ad5 remodeling of the cellular proteome. Furthermore, our integrative approach may be used as a model that can be adapted to address related questions involving other viruses and specific viral proteins that impact the cellular proteome during infection. We envisage that our findings will motivate future research to establish functional relationships underlying the dynamic association of host factors with viral genomes and the relevance of these virus-host interactions to mechanisms of host restriction.

## MATERIALS AND METHODS

**Cell culture.** A549 and U2OS cells were originally purchased from the American Type Culture Collection (ATCC; CCL-185 and HTB-96). A549 cells were maintained in Ham's F-12K medium (Gibco; 21127-022). U2OS cells were maintained in Dulbecco's modified Eagle's medium (DMEM; Corning; 10-013-CV). W162 cells (92) were a gift from G. Ketner (Johns Hopkins University) and maintained in DMEM. All cells were cultured at 37°C and 5% $CO_2$. All media were supplemented with 10% (vol/vol) fetal bovine serum (FBS) (VWR; 89510-186) and 1% (vol/vol) penicillin-streptomycin (Pen-Strep; 10,000 U/ml; Gibco; 15140-148).

**Viruses and infection.** Wild-type Ad5 was initially purchased from ATCC. The Ad5 E1B55K deletion mutant *dl*110 (93) and E4 deletion mutant *dl*1004 (11, 12) were gifts from G. Ketner (Johns Hopkins University). The E4orf6 deletion mutant *dl*355 (11) was a gift from D. Ornelles (Wake Forest University). Ad5 WT and *dl*110 virus strains were propagated on HEK293 cells. The *dl*1004 and *dl*355 mutant viruses were propagated on W162 cells. Virus particles were purified via two successive rounds of ultracentrifugation using a cesium chloride gradient. Virus particles were stored in 40% (vol/vol) glycerol at −80°C (long-term) and −20°C (short-term). Stock titers of viruses were determined by plaque assay on the respective cell lines used for propagation. All whole-cell proteome assays, immunoblotting, viral DNA replication by qPCR, and plaque assays were performed at a multiplicity of infection (MOI) of 10. All iPOND and immunofluorescence microscopy experiments were performed at an MOI of 40. Infections were performed in low volume on cell monolayers using respective media containing 2% FBS. After 2 h viral adsorption at 37°C, viral inoculum was aspirated and replaced with full serum growth medium (10% FBS) for the duration of the experiment. For infections with neddylation inhibition, MLN4924 (Sigma-Aldrich; 505477) was dissolved in dimethyl sulfoxide (DMSO) at 1 mM, used at 3 $\mu$M, and added to infected cell medium at 8 hpi. DMSO at an identical volume was added to a paired control set of infected cells.

**Antibodies.** The following primary antibodies were used for immunoblot analysis and immunofluorescence microscopy: antibodies against cellular proteins and BrdU, CUL5 (Bethyl; A302-173A), DCAF1 (Proteintech; 11612-1-AP), glyceraldehyde3-phosphate dehydrogenase (GAPDH) (GeneTex; GTX100118), HELLS/SMARCA6 (Proteintech; 11955-1-AP), MRE11 (Novus Biologicals; NB100-142), RAD50 (GeneTex; GTX70228), RECQL4 (Abcam; ab188125), SMARCAD1 (Sigma-Aldrich; SAB1407755), SMC5 (Proteintech; 14178-1-AP), SMC6 (GeneTex; GTX116832), SQSTM1 (Proteintech; 18420-1-AP), tubulin (Santa Cruz Biotechnology; sc-69969), and BrdU (Abcam; ab6326); antibodies against viral proteins, rabbit polyclonal against hexon, penton, and fiber (gift from J. Wilson), mouse anti-DBP (gift from A. Levine; clone B6-8), mouse anti-E1B55K (gift from A. Levine; clone 58K2A6), mouse anti-E4orf6 (gift from D. Ornelles; clone RSA#3). For immunoblot analysis, horseradish peroxidase (HRP)-conjugated secondary antibodies were purchased from Jackson Laboratories. For immunofluorescence, Alexa Fluor-conjugated secondary antibodies were purchased from Life Technologies.

**siRNA transfection.** The following siRNAs were obtained from Dharmacon: nontargeting control no. 1 (D-001810-01), SMARCAD1 (L-013801-00-0005), SMC5 (L-014117-01-0005), and SMC6 (L-018408-01-0005). siRNA transfections were performed following the standard protocol for Lipofectamine RNAiMAX (Invitrogen).

**Immunoblot analysis.** Samples were lysed directly in 1× NuPAGE lithium dodecyl sulfate sample buffer (Thermo Scientific; NP0007) supplemented with 100 mM dithiothreitol. Samples were boiled at 95°C for 10 min. Samples were separated by SDS-PAGE in morpholinepropanesulfonic acid (MOPS) buffer and transferred onto polyvinylidene difluoride (PVDF) membranes. Membranes were blocked in 5% (wt/vol) nonfat dry milk in Tris-buffered saline supplemented with 0.1% Tween 20 (TBS-T). Membranes were incubated with primary antibodies in milk (TBS-T) either for 1 to 2 h at room temperature or overnight at 4°C, washed thrice in TBS-T, incubated with HRP-conjugated secondary antibodies for 1 h at room temperature, and washed thrice in TBS-T. Proteins were visualized using Pierce ECL, West Pico, and West Femto Western blotting substrates (Thermo Scientific) and detected with a Syngene G-Box. Immunoblots were processed and quantified by densitometry of pixels in FIJI ImageJ (v1.52p).

**Immunofluorescence imaging.** A549 cells were seeded onto glass coverslips in 24-well plates and either mock infected or infected with wild-type Ad5 or *dl*1004. At 22 hpi, BrdU labeling was performed with the addition of 10 $\mu$M BrdU to the medium, and cells were incubated for 2 h at 37°C and then washed thrice in phosphate-buffered saline (PBS) All cell monolayers were fixed at 24 hpi in 4% (wt/vol) paraformaldehyde in PBS for 15 min and washed thrice in PBS. Cell monolayers were permeabilized with 0.5% Triton X-100 in PBS for 10 min and washed thrice in PBS. DNA hydrolysis was performed on BrdU-labeled cell monolayers upon incubation with 70 mM NaOH for 30 min at room temperature. Samples were neutralized with 100 mM sodium borate, pH 8.5, for 30 min at room temperature and washed

thrice in PBS. All samples were blocked in 3% (wt/vol) bovine serum albumin (BSA) in PBS for 1 h at room temperature, incubated with primary antibodies in this blocking solution for 1 h at room temperature, and washed thrice with blocking solution. Samples were incubated with Alexa Fluor-conjugated secondary antibodies (Invitrogen; Alexa Fluor 488 and 568 as anti-mouse or anti-rabbit) and 1 $\mu$g/ml 4,6-diamidino-2-phenylindole (DAPI) for 1 h at room temperature, washed thrice in PBS, and mounted onto glass sides with ProLong Gold antifade reagent (Cell Signaling Technologies). Images were acquired on a Zeiss LSM 710 confocal microscope (University of Pennsylvania Cell and Development Microscopy Core) using Zen 2011 software. Images were processed in FIJI ImageJ (v1.52p).

**Viral DNA quantification by qPCR.** At 48 post-transfection with siRNA reagents, cells were infected as indicated and harvested by trypsinization and pelleting. Total DNA was isolated via the PureLink genomic DNA kit (Invitrogen). qPCR was performed with primers for the Ad5 DBP genomic region (forward primer, 5'-ATCACCACCGTCAGTGAA-3'; reverse primer, 5'-GTGTTATTGCTGGGCGA-3') and normalized to cellular tubulin (forward primer, 5'-CCAGATGCCAAGTGACAAGAC-3', and reverse primer, 5'-GAGTGAGTGACAAGAGAAGCC-3'). For each virus strain infection series, samples were assessed at and further normalized to the 4-hpi time point of siControl using the threshold cycle ($\Delta\Delta C_T$) method via SYBR green PCR master mix (Life Technologies). qPCR was performed on a QuantStudio 7 Flex real-time PCR system (Applied Biosystems).

**Plaque assay.** Cells were infected as indicated and harvested by scraping and pelleting at the indicated times postinfection. Samples were frozen and thawed for three successive cycles using liquid nitrogen and a 37°C water bath. Centrifugation at maximum speed (21,130 × $g$) for 5 min at 4°C was performed to remove cell debris. Lysates were serially diluted in DMEM containing 2% (vol/vol) FBS and 1% (vol/vol) Pen-Strep to infect confluent monolayers of W162 cells seeded in 12-well plates. After 2 h virus adsorption at 37°C, infection medium was aspirated and cells were overlaid with DMEM containing 0.45% (wt/vol) SeaPlaque agarose (Lonza), 2% (vol/vol) FBS, and 1% (vol/vol) Pen-Strep. After 7 days postinfection, 10% (wt/vol) trichloroacetic acid was added to each well for 30 min at room temperature. Medium was aspirated, and plaques were stained with 1% (wt/vol) crystal violet in 70% (vol/vol) methanol.

**iPOND sample preparation and mass spectrometry analysis.** The iPOND assay was performed as previously described (45, 55) with adaptations to incorporate virus infection. To compare iPOND-enriched proteins during AdΔE4 infection to those during Ad5 WT infection, U2OS cells were used as previously described (45). For each condition, eight 15-cm dishes of 90% confluent U2OS cells (~1.8 × 10⁷ cells) were either mock infected or infected with Ad5 WT or AdΔE4 (dl1004) at an MOI of 40. Three independent biological replicate experiments were performed per condition. After 2 h virus adsorption at 37°C, viral inoculum was supplemented with additional full culture medium (DMEM supplemented with 10% [vol/vol] FBS and 1% [vol/vol] Pen-Strep) and incubated at 37°C for 24 h. The samples were pulsed with 10 $\mu$M EdU (Invitrogen) for 15 min at 37°C and then fixed with 1% paraformaldehyde in PBS for 20 min at room temperature. The cross-linking reaction was quenched with 125 mM glycine, and cells were harvested by scraping and pelleting. Per condition, samples were divided into two pellets for subsequent iPOND processing as previously described (44, 55), with adaptations. Following the click chemistry reaction, cell pellets were resuspended in 0.5 ml lysis buffer (20 mM HEPES, pH 7.9, 400 mM NaCl, 0.5% Triton X-100, 10% glycerol, 1 mM EDTA, 1 mM dithiothreitol [DTT], 1× cOmplete protease inhibitor [Roche], and 1 mM phenylmethylsulfonyl fluoride). Samples were sonicated with a Bioruptor (Diagenode) for 20 min using 30-s on/off cycles at the highest intensity. Samples were centrifuged at maximum speed to remove cell debris. Streptavidin magnetic beads (Invitrogen: Dynabeads M-280) were used to isolate DNA-protein complexes by incubating the lysates with the beads for 16 to 18 h at 4°C with end-over-end rotation. Beads were washed with lysis buffer, followed by one wash with 1 M NaCl, four washes with wash buffer (20 mM HEPES, pH 7.4, 110 mM potassium acetate [KOAc], 2 mM MgCl₂, 0.1% Tween 20, 0.1% Triton X-100,150 mM NaCl), and one wash with PBS. Isolations from the same condition were combined by sequential elution in a total of 60 $\mu$l of 1× LDS sample buffer (Invitrogen) containing 100 mM DTT and boiling at 95°C for 10 min. To reverse the cross-links, eluates were further boiled at 95°C for 45 min.

For mass spectrometry analysis, iPOND isolates were separated on 10% Bis-Tris Novex minigels (Invitrogen) using MOPS buffer. The gels were stained with Coomassie blue and cut into four 2- by 7-mm gel segments. Gel pieces were destained with 50% methanol and 1.25% acetic acid, reduced with 5 mM DTT for 1 h, and alkylated with 40 mM iodoacetamide (Sigma-Aldrich) for 45 min, both at room temperature. Gel pieces were incubated with 20 mM ammonium bicarbonate and rapidly dehydrated with acetonitrile. To digest the proteins, trypsin was added to the gel pieces (5 ng/$\mu$l in ammonium bicarbonate) and incubated overnight at 37°C. Peptides were extracted using the sequential addition of 0.3% trifluoroacetic acid (TFA) and then 50% acetonitrile; the two solutions were then combined for each condition. Tryptic digests were analyzed by nano-liquid chromatography coupled online with tandem mass spectrometry (nLC-MS/MS) on a nanoLC Ultra (Eksigent Technologies) coupled to a hybrid LTQ Orbitrap Elite mass spectrometer (Thermo Fisher Scientific). Peptides were separated via reversed-phase chromatography on a nanocapillary column (75-$\mu$m inside diameter [i.d.] by 15-cm C₁₈ Reprosil-pur 3 $\mu$m, 120 Å) in a Nanoflex chip system. The mobile phase A contained 1% methanol and 0.1% formic acid, and mobile phase B contained 1% methanol, 0.1% formic acid, and 80% acetonitrile. Peptides were separated at a flow rate of 300 nl/min along a 90-min gradient from 5 to 35% B. For data-dependent MS/MS scans, the full scan range was performed in the Orbitrap and set to 300 to 1,800 $m/z$ at a 240,000 resolution. The top-20 most abundant precursor ions at every duty cycle were selected for fragmentation in the ion trap, with a minimum signal of 1,500 counts, dynamic exclusion enabled with a repeat count of 1, exclusion window of 500 and duration of 60 s, isolation width of 2 $m/z$, normalized

collision energy of 35, and injection waveform enabled. Fourier transform mass spectroscopy (FTMS) full-scan automatic gain control (AGC) target was set to 1E6, while MS/MS AGC target was set to 1E4. FTMS full-scan maximum fill time was 500 ms, while the ion trap MS/MS fill time was 50 ms. Microscans were set at one.

The raw mass spectrometer files were processed for protein identification using the Proteome Discoverer software (v2.4; Thermo Scientific) and the Sequest HT algorithm with a peptide mass tolerance of 10 ppm, fragment $m/z$ tolerance of 0.25 Da, and a false-discovery rate (FDR) of 1% for proteins and peptides. Quantification was performed using a label-free approach using the "Precursor ions quantifier" node, and peptide abundances were rolled up into protein abundance using the summed abundance algorithm using only unique or razor peptides. All peak lists were searched against the UniProtKB/Swiss-Prot database of Human (9606; downloaded February 2021; 20,397 entries) and Adenovirus C serotype 5 (28285; downloaded February 2021; 31 entries) sequences using the parameters as follows: enzyme, trypsin; maximum missed cleavages, 2; fixed modification, carbamidomethylation (C); variable modifications, oxidation (M) and protein N-terminus acetylation.

**WCP sample preparation and mass spectrometry analysis.** In biological quadruplicate, A549 cells were mock infected or infected with Ad5 WT at 90% confluence in 60-mm dishes at an MOI of 10. Cell monolayers were scraped and collected at 16, 24, and 48 hpi. Mock-infected samples were scraped and collected at 24 hpi. Samples were pelleted, washed with PBS, flash frozen in liquid nitrogen, and stored at −80°C until further processing. Cell pellets were lysed in ice-cold lysis buffer (8 M urea in 50 mM ammonium bicarbonate, pH 8.0) with Halt protease inhibitor cocktail (Thermo Fisher Scientific) by vortexing, following sonication with a Bioruptor (Diagenode) on medium power (200 W) for 30 s on/off for 20 cycles. Samples were then centrifuged for 10 min, 17,000 × $g$, at 4°C, and the supernatant was transferred to a new set of tubes. Samples were reduced using 5 mM DTT for 1 h at room temperature and alkylated with 10 mM iodoacetamide (IAA) in the dark for 45 min at room temperature. Samples were diluted to 1.5 M urea with 50 mM ammonium bicarbonate, pH 8.0, and then digested with trypsin (Promega) at an enzyme-to-substrate ratio of ~1:50 for 12 h at room temperature. Peptides were loaded onto a Hamilton $C_{18}$ stage tip, and high-pH reverse-phase fractionation was performed using 8, 12, 18, 24, 30, 36, 42, and 60% acetonitrile with 100 mM ammonium formate buffer (pH 10.0) to bin peptides by hydrophobicity (94). Fraction 1 was then combined with fraction 5, i.e., 8% and 30%, and 2 with 6, etc., to obtain four final fractions for MS analysis. Samples were desalted using P200 columns with a $C_{18}$ 3M plug (3M Bioanalytical Technologies) and reconstituted in 0.1% trifluoroacetic acid (TFA) prior to nLC-MS/MS analysis.

Peptide quantification by nLC-MS/MS was performed on a Thermo Fisher Ultimate 3000 Dionex liquid chromatography system and a Thermo Fusion mass spectrometer. The mobile phase A contained 0.1% formic acid, and mobile phase B contained 0.1% formic acid and 80% acetonitrile. Peptides were separated at a flow rate of 300 nl/min along a 120-min gradient from 2 to 50% mobile phase B. Samples were quantified by $A_{280}$ absorbance, and 1 $\mu$g of each was injected onto a laser-pulled 2.4-$\mu$m $C_{18}$ particle size silica column of 35 cm in length. Samples were run with MS1 settings of 300 to 1,200 $m/z$ window, a resolution of 60,000, normalized AGC target of 125%, and maximum inject time (MIT) of 54 ms. Fragment (MS2) scans were collected in data-dependent mode with a TopN loop count of 10, resolution of 30,000, normalized AGC target of 150%, and MIT of 100 ms. Fragmentation was performed with high-energy collisional dissociation (HCD) using normalized collision energies (NCE) of 29%.

The raw mass spectrometry files were processed for protein identification using the Proteome Discoverer software (v2.4; Thermo Scientific) and the Sequest HT algorithm with 10-ppm tolerance for MS1 and 0.02-Da tolerance for fragment masses. The multiple fractions from the same sample were imported into the Proteome Discoverer search as "fractions." Therefore, the fractions were merged in the search as a single sample. Thus, a unique protein abundance was quantified within the respective biological replicate for each sample, regardless of which fraction contained the protein. All peak lists were searched against the UniProtKB/Swiss-Prot database of Human proteins (9606; downloaded February 2021; 20,397 entries) (95) and an Ad5 viral proteome that was curated from a reannotated Ad5 genome (96).

An FDR cutoff of 1% was used for all peptide spectral matches using Percolator, and a subsequent 1% FDR for all protein IDs with peptides less than a $q$ value of 0.01 was selected (97). To match features between runs, we used a Minora feature detector at tolerances of 3 ppm to identify peptides and a maximum retention time shift of 2 min.

**Proteomics data analysis.** Proteomics data analysis was performed using in-house scripts written in the R software environment (98). Protein abundances determined by Proteome Discoverer were used for the proteomics protein quantification data analysis. Abundance values of "0" were reassigned as "NA." Protein quantifications were $log_2$ transformed and normalized by biological replicate sample median. To assess protein abundance changes across compared conditions, $log_2$ fold change was calculated based on the mean $log_2$-transformed, normalized protein abundance values for respective conditions. $P$ values were calculated using unpaired, two-tailed Student $t$ tests comparing $log_2$-transformed, normalized proteins abundance values. Multiple test correction was not performed (99). Relative abundance was measured by statistical z-score transformation of the average $log_2$-transformed, normalized abundance values within each condition.

For the iPOND data analysis, a specific protein was defined as enriched on the mock, Ad5 WT, or AdΔE4 genome by fulfilling one of three conditions: (i) if the protein was quantified in at least 2 biological replicates for both infections, enrichment was defined by a $log_2$ fold change of ≥0.585 and $P < 0.05$; (ii) if the protein was quantified in at least 2 biological replicates for one infection but only 1 biological replicate for the compared infection, enrichment was defined by a $log_2$ fold change of ≥0.585; and (iii) if the protein was quantified in at least 2 biological replicates for one infection and 0 biological replicates

for the compared infection, the protein was defined as "uniquely" associated with the genome for which the protein was quantified in ≥2 replicates. A specific protein was defined as unenriched for a specific infection if it was identified in at least 2 biological replicates for that condition but did not meet the definition for enrichment as described above.

In order to identify changes of protein relative abundance in response to Ad5 WT infection, protein quantifications in the WCP data set were compared for each infection time point with respect to mock infection. A specific protein was defined as increased or decreased for a time point of infection, compared to mock, by fulfilling one of three conditions: (i) if the protein was quantified in at least 2 biological replicates for mock and the infection time point, increase/decrease was defined by a $\log_2$ fold change of ≥0.585 and $P < 0.05$; (ii) if the protein was quantified in at least 3 biological replicates for mock or the infection time point but only 1 biological replicate for the compared infection, increase/decrease was defined by a $\log_2$ fold change of ≥0.585; (iii) if the protein was quantified in at least 3 biological replicates for mock or the infection time point and 0 biological replicates for the compared infection, the protein was defined as "uniquely" identified in the infection for which the protein was quantified in ≥3 biological replicates. A specific protein was defined as unchanged for a time point of infection compared to mock if it was identified in at least 2 biological replicates for each condition but the absolute value of the $\log_2$ fold change was <0.585 or the $P$ value was ≥0.05.

**Gene Ontology and pathway analysis.** Gene Ontology (GO) and pathway analysis was performed using the STRING protein-protein interaction database functional enrichment annotations within the Cytoscape network visualization software package (100, 101). The STRING-annotated enrichment terms were filtered at 50% redundancy, using the in-built STRING algorithm. Enrichment terms with FDR of <0.05 were considered significant and are presented as $-\log_{10}$-transformed values for visualization purposes.

**Data availability.** Raw MS files associated with this work have been deposited to the public database the ProteomeXchange Consortium repository via the PRIDE partner repository (102). The accession number for the MS data is PXD025339.

## SUPPLEMENTAL MATERIAL

Supplemental material is available online only.

**FIG S1**, PDF file, 1 MB.
**FIG S2**, PDF file, 0.1 MB.
**FIG S3**, PDF file, 0.5 MB.
**FIG S4**, TIF file, 1.9 MB.
**FIG S5**, TIF file, 2.9 MB.
**FIG S6**, PDF file, 0.4 MB.
**TABLE S1**, XLSX file, 7.2 MB.
**TABLE S2**, XLSX file, 1.6 MB.

## ACKNOWLEDGMENTS

We thank all members of the Weitzman lab for critical discussions, input, and review of the manuscript. We are grateful to G. Ketner, D. Ornelles, A. Levine, and J. Wilson for the reagent gifts. We also thank the University of Pennsylvania Cell and Developmental Biology Microscopy Core for imaging assistance.

Work was supported through NIH grants R01-AI145266, R01-AI121321, R01-CA097093, and R21-AI157416 (M.D.W.), as well as R01-AI118891 and P01-CA196539 (B.A.G.). Additional support was obtained through Individual National Research Service Awards F32-AI147587 (J.M.D.) and F32-AI138432 (A.M.P.), T32 Training Grant in Virology T32-AI007324 (K.K.L.), T32 Training Grant in Cell and Molecular Biology T32-GM007229 (N.G.), and an Academic Diversity Postdoctoral Fellowship from the Office of the Vice Provost for Research at the University of Pennsylvania (E.D.R.).

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
