## [Reviewer comments · mSystems]

Adenovirus remodeling of the host proteome and of host factors associating with viral genomes

Joseph Dybas, Krystal Lum, Katarzyna Kulej, Emigdio Reyes, Richard Lauman, Matthew Charman, Caitlin Purman, Robert Steinbock, Nicholas Grams, Alex Price, Lydia Mendoza, Benjamin Garcia, and Matthew Weitzman

Corresponding Author(s): Matthew Weitzman, University of Pennsylvania Perelman School of Medicine

Review Timeline:

Submission Date:	April 15, 2021
Editorial Decision:	May 27, 2021
Revision Received:	August 7, 2021
Accepted:	August 9, 2021

Editor: Jack Gilbert

Reviewer(s): The reviewers have opted to remain anonymous.

Transaction Report:

DOI: <https://doi.org/10.1128/mSystems.00468-21>

May 12, 2021

Dr. Matthew D. Weitzman
University of Pennsylvania Perelman School of Medicine
Department of Pathology and Laboratory Medicine
4050 Colket Translational Research Building
3501 Civic Center Blvd
Philadelphia, PA 19104-4318

Re: mSystems00468-21 (Adenovirus remodeling of the host proteome and of host factors associating with viral genomes)

Dear Dr. Matthew D. Weitzman:

Thank you for submitting your manuscript to mSystems. We have completed our review and I am pleased to inform you that, in principle, we expect to accept it for publication in mSystems. However, acceptance will not be final until you have adequately addressed the reviewer comments.

Thank you for the privilege of reviewing your work. Below you will find instructions from the mSystemseitorial office and comments generated during the review.

Preparing Revision Guidelines

For complete guidelines on revision requirements, please see the Instructions to Authors at [link to page]. **Submissions of a paper that does not conform to mSystems guidelines will delay acceptance of your manuscript.**

Sincerely,

Frank Schmidt

Editor, mSystems

Journals Department
Reviewer comments:

Reviewer #2 (Comments for the Author):

Dybas et al. study the impact of AdV5 infection on the host cell proteome at late stages of the infection cycle. They use an elegant two-armed approach, in which they determine kinetics of whole cell proteomic changes and in parallel profile the proteome on replicated AdV DNA. More specifically, they focus on differences observed in presence and absence of the viral E4 ORF, which contains several modulators of transcription, cell-cycle, and DNA repair. In the whole cell proteomics approach, they reveal distinct modulation of host protein expression. In addition to corroboration of previously published host proteins targeted by the viral E4 ORF, additional proteins were addressed, which decreased upon infection. These proteins are associated with chromosomal organization. The authors suggest that the observed degradation of the host proteins upon infection was mediated by the viral E3 ubiquitin-ligase E1B55K/E4orf6. This is to some extent supported by the presented western blots and pharmacological inhibitor experiments.

To further investigate the effects of gene products from the viral E4 ORF on host proteins associating to viral genomes, iPOND analysis were performed. The comparison of WT AdV5 genomes and Δ E4 genomes revealed an expected increase of E4 ORF target proteins involved host antiviral responses upon E4 ORF deletion. This confirms that the E4 ORF leads to degradation of specific host proteins. GO analysis indicated that the E4 ORF leads to degradation of proteins involved in "DNA metabolic process" and "chromosome organization". Among these host proteins, the authors characterized SWI/SNF and SMC proteins. Using immunofluorescence staining and siRNA mediated silencing, the authors propose that SWI/SNF and SMC proteins have a negative impact on AdV5 replication and this is antagonized by the viral E4 ORF.

The data is presented in a clear and easily accessible manner and largely supports the claims made. The restoration of protein expression in Δ E4 infected cells is not clearly shown and may require some additional attention. In addition, the writing sometimes lacks flow, at times is repetitive and a better guidance of the reader through the manuscript should be considered. Overall, this is a nicely designed and executed study, which requires only minor modifications.

Minor comments:

General

- Writing lacks flow, repetitive words and hard to follow at times (e.g. lines 313ff)

- Lack of mechanistic explanation of viral target protein functions in results - hard to follow the rationale
- Nice materials and methods
- Clean figures

Figure 1

- Nice schematic representation, very clear and clean

Figure 2

- B) Please comment why the number of total proteins is increased 48hpi.
- E) Explain what the violet frame (proteins in the violet frame) signifies.

Figure 3

- A) Redundant to Fig. 2C. Are these plots really discussed in the text?
- B) Color scheme not optimal. In the current figure, it looks like there is little change at 16 h and 24 h. Possibly adjust color scheme to better visualize the early changes.
- C) Text part: Why were these four hits specifically followed up on and why are they relevant?

Figure 4

- Nice blots
- C) For the partial rescue a quantification of the protein bands will be helpful.
 - o It is hard to see the partial rescue for some hits (RECQL4, SMARCA6, SQSTM1, DCAF1)

Figure 5

- B) It seems counterintuitive that the number of interaction partners is not decreased in the Δ E4 sample. Please comment. Possibly the changes are only quantitative and not qualitative and hence cannot be seen here.
- E) Lack of discussion why this process is / these targets are chosen for further analysis

Figure 6

- B) Is that phenotype seen in all infected cells? Possibly show an overview frame with several cells in addition.
- B) Is there another marker for viral replication centers? Viral DNA / PMA bodies?
- B) In the text it would be beneficial to give more information on the two host proteins. Also please comment on the perinuclear staining of SMC6.

Figure 7

- Why was there a change from SMC6 to SMC5 between Fig 6 and here? Could you also show immunofluorescence of SMC5 in Fig. 5?
- Difficult to see the different lines in the different shades of grey. Maybe change colors.
- How can the different effects be explained for the three targets? Why does siSMARCA1 not increase viral genomes in Δ E4? Please clarify.

Reviewer #3 (Comments for the Author):

The authors present a detailed analysis of the changing cellular proteome during adenovirus infection, and explore specifically the changes in protein association with the viral genome that arise as a result of deletion of the viral E4 gene, which is known to encode several proteins that

target cellular proteins for degradation. Their overall conclusions are (1) that there are several kinetic classes of protein that are depleted from infected cells in an E4-dependant manner, and (2) that E4 proteins cause loss of proteins from the SW/SNF and SMC families that otherwise associate with replicating viral DNA; they further show that these proteins are inhibitory to the productivity of infection. Overall, there is considerable novelty in the study, particularly including the use of iPOND to look at proteins associated with the viral DNA, which will be of substantial interest to fellow scientists in the virology community.

The manuscript is clearly written and the data presentation is of very high standard. The data support their conclusions, in most respects, very well. I list some points that need to be considered below.

Fig 3A, B and Line 179-192: It seems logical to assume that the three temporal classes of cellular protein degradation targets have been made mutually exclusive, ie proteins that are reduced of mock at 16 hr are still reduced of the mock at 24 and 48 hr but they have been subtracted out from the proteins found to be reduced at these later time points to create the lists for GO analysis. Is that correct? If so, the text needs to be improved to make that clear.

Fig 4C, D and Line 237: This conclusion is hard to justify without detailed quantitation of the relevant protein bands, with normalization to the control, across multiple biologic repeats. By eye, any difference between wt and mutant in the extent of loss of the newly identified targets (RECQL4, SMARCA6, SQSTM1, DCAF1) seems minimal at best, especially when compared to the complete protection from degradation by each of the mutations of the well-established degradation targets RAD50 and MRE11. The authors do go on (line 240-3) to discuss this difference, but the primary conclusion seems overstated from the data. These are indeed E4-dependent degradation targets but presenting them primarily as targets of the E1B/E4 Orf6 Ub-ligase is too strong.

Line 265 and following. It's unclear why the work switched from A549 (a well-established host system for Ad5) to U2OS cells. The latter are less infectable by the virus, which the authors clearly know as they increased the moi for the iPOND series of experiments in U2OS to 40 rather than the 10 used for the A549 work, which should achieve approx. similar levels of infection. The difference in cell line does though make bringing the two sets of data together rather difficult as the stage of infection at a given time p.i. cannot be assumed to be the same.

Fig 5C and Fig S2D, line 290-307. The replicating DNA that is the source of the protein for MS proteomics will be a mix of viral and cell DNA. In order to draw the conclusions that are offered about differences in specific protein enrichment between Ad5wt and delE4, it needs to be demonstrated that the proportion of the labelled (replicating) DNA that is viral is similar in each case. Otherwise a difference between the two could be down to a greater abundance of viral DNA in one set of samples.

Line 603-5. How were the data from each of the four gel slices from a sample handled subsequent to collection? At what point, if at all, were they brought back together? What happens to a protein that is in fraction 2 from one replicate and - being at the margin - ends up in fraction 3 from the other replicates? Given the declared aim of the fractionation is to improve discovery rates, the measured abundance of a protein might change simply for moving from one fraction to another. Also, how was the quantitation normalized across the four fractions taken from a sample?

Line 670. Were the data from the four peptide bins for each sample combined together before doing the protein discovery analysis? If not, differences at the fractionation margins might give rise to a peptide 'moving' between fractions and so not giving a positive detection in the required 3 of 4 replicates.

Some of the cited refs are not complete (at a glance, this includes 83, 84, 85 and 87).

We thank the editor and reviewers for the constructive comments and complimentary statements about our study design, experimental execution, and data presentation. As detailed in our responses below, we believe our responses have substantially improved the manuscript. We have extensively revised the text to expand the experimental rationale, improve the writing flow, and limit repetition. To more clearly show levels of protein restoration in $\Delta E4$, $\Delta E4orf6$, and $\Delta E1B55K$ infected cells, we have also quantified the protein expression in cells infected with Ad5 WT and the respective mutant strains by densitometry analysis. By confocal microscopy, we provide overview fields of view of infected cells and an additional marker to visualize viral replication centers as evidence of proteins associating with viral genomes. We believe our additional data and text changes lend support to our manuscript describing Ad5 E4-dependent manipulations of the host proteome during infection. Below we have included our point-by-point response to each reviewer comment. Our responses are marked with ">" and relevant sections of the manuscript are highlighted in yellow.

Reviewer #2 (Comments for the Author):

Dybas et al. study the impact of AdV5 infection on the host cell proteome at late stages of the infection cycle. They use an elegant two-armed approach, in which they determine kinetics of whole cell proteomic changes and in parallel profile the proteome on replicated AdV DNA. More specifically, they focus on differences observed in presence and absence of the viral E4 ORF, which contains several modulators of transcription, cell-cycle, and DNA repair. In the whole cell proteomics approach, they reveal distinct modulation of host protein expression. In addition to corroboration of previously published host proteins targeted by the viral E4 ORF, additional proteins were addressed, which decreased upon infection. These proteins are associated with chromosomal organization. The authors suggest that the observed degradation of the host proteins upon infection was mediated by the viral E3 ubiquitin-ligase E1B55K/E4orf6. This is to some extent supported by the presented western blots and pharmacological inhibitor experiments.

To further investigate the effects of gene products from the viral E4 ORF on host proteins associating to viral genomes, iPOND analysis were performed. The comparison of WT AdV5 genomes and $\Delta E4$ genomes revealed an expected increase of E4 ORF target proteins involved host antiviral responses upon E4 ORF deletion. This confirms that the E4 ORF leads to degradation of specific host proteins. GO analysis indicated that the E4 ORF leads to degradation of proteins involved in "DNA metabolic process" and "chromosome organization". Among these host proteins, the authors characterized SWI/SNF and SMC proteins. Using immunofluorescence staining and siRNA mediated silencing, the authors propose that SWI/SNF and SMC proteins have a negative impact on AdV5 replication and this is antagonized by the viral E4 ORF.

The data is presented in a clear and easily accessible manner and largely supports the claims made. The restoration of protein expression in $\Delta E4$ infected cells is not clearly shown and may require some additional attention. In addition, the writing sometimes lacks flow, at times is repetitive and a better guidance of the reader through the manuscript should be considered. Overall, this is a nicely designed and executed study, which requires only minor modifications.

Minor comments:

General

- Writing lacks flow, repetitive words and hard to follow at times (e.g. lines 313ff)

>We have extensively revised the text to improve the flow of writing and limit repetition.

- Lack of mechanistic explanation of viral target protein functions in results - hard to follow the rationale

>The text now includes an expanded description of the proteins in the WCP and iPOND results sections to provide more rationale for why we chose to perform experimental follow-up on the selected proteins. We now include more explanation of the importance of the WCP target proteins (HELLS/SMARCA6, RECQL4, DCAF1, SQSTM1/p62) (Lines 187-210). For the iPOND targets (SMARCAD1 and SMC5/6) we describe the selection on the basis of GO analysis and proteomics quantification data in the Results section (Lines 315-328). We further expand on the known functions of these proteins and how they might be relevant for viral infection in the Discussion section (Lines 458-491).

- Nice materials and methods

>We thank the reviewer for this complimentary statement.

- Clean figures

>We thank the reviewer for this complimentary statement.

Figure 1

- Nice schematic representation, very clear and clean

>We thank the reviewer for this complimentary statement.

Figure 2

- B) Please comment why the number of total proteins is increased 48hpi.

>The Reviewer has identified an interesting characteristic of the data. At 48hr there are fewer proteins identified overall, compared to mock, 16hpi or 24hpi. Furthermore, there are slightly more proteins identified in <2 replicates at 48hpi, compared with mock, 16 or 24hpi. Additionally, there is less overlap in the proteins that are identified during mock and 48hpi infection, as measured by comparing the unions of the proteins identified in these two conditions. This suggests that many of the identified proteins are at relatively lower abundance, which is manifested in the large number of decreased proteins shown in Figure 2B. However, this characteristic of the data also leads to many “unchanged” proteins to be calculated between mock and 48hpi, based on the original parameters of analysis, which defined as “unchanged” any protein that was not reproducibly quantified as increased or decreased. Based on the reviewer’s comment, we have reconsidered how best to calculate the proteins that are unchanged between two conditions. We have set criteria to define a protein as “unchanged” to require the protein in question to be identified in at least 2 biological replicates in each of the compared conditions, in addition to not meeting the fold change and p-value requirements for a significant increase or decrease in abundance. In this way, we are better categorizing those proteins that are confidently quantified in both conditions and measured as unchanged in abundance. Performing the calculation in this way results in total numbers of increased, decreased, and unchanged proteins that better reflect the relative numbers of identified proteins at each infection sample. The method for categorizing the “increased” and “decreased” proteins has not changed since these calculations already considered biological replicate numbers. The text in the Methods section (Line 735-737) has been modified to describe method by which we categorized the “unchanged” proteins and Figure 2B has also been changed.

- E) Explain what the violet frame (proteins in the violet frame) signifies.

>The proteins listed in the violet frames on the volcano plots in Figure 2E represent proteins that were “uniquely” identified in the mock (uninfected) condition and were not identified at the matched infection time point. Specifically, LIG4 was identified in 3 of 4 biological replicates for the mock condition, but in 0 replicates from the 16, 24, or 48hpi infection time points. Similarly, DAXX was identified in 4 of 4 biological replicates in the mock condition, but in 0 replicates of the 48hpi time point. Since there is no abundance for these “unique” proteins during infection, a fold change and corresponding p-value cannot be calculated as a point on the volcano plot. However, the decrease from mock to infection is important to note for these known substrates of the E1B55K/E4orf6 ligase. Therefore, they were included in the violet frame. The Results (Lines 158-160) and Figure 2 legend (Lines 796-800) have been revised to clarify these “unique” proteins.

Figure 3

- A) Redundant to Fig. 2C. Are these plots really discussed in the text?

>The plots referred to in Figure 3A are a different representation of the data shown in Figure 2C. The Venn diagrams in Figure 2C show the number and intersection of the decreased proteins at each time point. The line plots in Figure 3A show the fold changes at each time point for the proteins categorized as decreased at 16, 24, or 48hpi of infection. We believe that the utility of including the data in Figure 3A is to show the overall dynamics of the proteins decreased at each time point. We have expanded the explanation of this figure in the Results section (Lines 170-174) to reflect this.

- B) Color scheme not optimal. In the current figure, it looks like there is little change at 16 h and 24 h. Possibly adjust color scheme to better visualize the early changes.

>We have adjusted the color scheme in response to the reviewer’s request. We note that each GO terms included in the heatmap was statistically significantly enriched at that time point (FDR < 0.05), regardless of color shade. Overall, there are more proteins decreased at 48hpi, compared to 16 or 24hpi, which increases the statistical significance of the enriched terms at 48hpi relative to 16hpi and 24hpi. We have revised the Results section (Lines 177-183) to specify this inclusion criteria for the GO terms and adjusted the coloring of Fig 3B.

- C) Text part: Why were these four hits specifically followed up on and why are they relevant?

>We have added more rationale to explain why the targets were selected for follow-up (Lines 187-210).

Figure 4

- Nice blots

>We thank the reviewer for this complimentary statement.

- C) For the partial rescue a quantification of the protein bands will be helpful. It is hard to see the partial rescue for some hits (RECQL4, SMARCA6, SQSTM1, DCAF1).

>The results shown for immunoblot analysis are representative of multiple biological replicates and recapitulate the abundance changes quantified in the mass spectrometry data. Additionally,

we have quantified the protein abundances in the blots by densitometry analysis (Figure S2). These analyses are described in the Results (Lines 237-240) and Methods (Lines 556) sections.

Figure 5

- B) It seems counterintuitive that the number of interaction partners is not decreased in the $\Delta E4$ sample. Please comment. Possibly the changes are only quantitative and not qualitative and hence cannot be seen here.

>We hypothesize that during wildtype infection the Ad5 E4 gene products impact host proteins associated with viral genomes, some of which may be mediated by either degradation of these proteins or mislocalizing them away from viral genomes. Indeed, it is known that E4 gene products such as E4orf6 and E4orf3 act to degrade and mislocalize host proteins such as the MRN complex and PML, which we detect localized to the viral genome in the $\Delta E4$ mutant infection. We therefore expect that the number of associating proteins will reflect a combination of active recruitment as well as targeting by early protein to prevent association. Our data show globally similar numbers of proteins enriched on the Ad5 wildtype (236 proteins) or $\Delta E4$ (264 proteins) genomes. We have revised the Results section to include a description of the two alternate scenarios for proteins being enriched on the viral genome in the absence of E4 and how the hypothesis relates to the data we show (Lines 276-289).

- E) Lack of discussion why this process is / these targets are chosen for further analysis

>As we described in our response above, the text now includes more rationale for why we chose SMARCAD1 and SMC5/6 proteins for further analysis. The GO term of “Chromosomal Organization” was enriched in the WCP decreased proteins and iPOND $\Delta E4$ associated proteins. We describe this basis of GO analysis and proteomics quantification in the Results section (Lines 315-328). We further expand on the known functions of these proteins and how they might be relevant for viral infection in the Discussion section (Lines 458-491).

Figure 6

- B) Is that phenotype seen in all infected cells? Possibly show an overview frame with several cells in addition.

>We have now included overview frames containing multiple cells (Figure S5). The immunofluorescence images show that relative to cells infected with Ad WT, a majority of cells infected with Ad $\Delta E4$ display enriched accumulations of SMARCAD1 and SMC6 at VRCs as marked by either DBP or BrdU (Figures 6B, S4, and S5). Our microscopy observations recapitulate the abundance changes quantified in our iPOND data.

- B) Is there another marker for viral replication centers? Viral DNA / PMA bodies?

>In addition to staining with the viral DNA-binding protein DBP, we have now included another marker for viral replication centers by labeling cells with the thymidine analog bromodeoxyuridine (BrdU), which becomes incorporated into newly replicated viral DNA (Lines 348-351, Line 355, Lines 361-362, Lines 559-566). We observe an accumulation of both SMARCAD1 and SMC6 at sites of newly replicated DNA in the absence of Ad E4 gene products (Figure S4). These findings are consistent with our observations that SMARCAD1 and SMC6 colocalize with DBP under similar conditions.

- B) In the text it would be beneficial to give more information on the two host proteins. Also please comment on the perinuclear staining of SMC6.

>We have added more information about these host proteins and their functions (Lines 458-491). We did not observe perinuclear staining of SMC6 in every cell in either the mock-infected or infected conditions. We propose that this localization is not an artifact, since a similar perinuclear localization was observed upon transfection of SMC6 (PMID 33092197) and occasionally in alternative commercial SMC6 antibody product sheets. The perinuclear staining may be suggestive of cell cycle differences in cells. As the perinuclear staining appeared to be at similar penetrance in each condition (Fig S5) we do not focus on its significance for infection. We have now indicated the perinuclear observation in the text (Lines 359-360).

Figure 7

- Why was there a change from SMC6 to SMC5 between Fig 6 and here? Could you also show immunofluorescence of SMC5 in Fig. 5?

>The original Figure 6B showed immunofluorescence to detect SMC6, while Figure 7 showed data for both SMC5 and SMC6. While SMC5 and SMC6 were both increased in fold change by iPOND (Figure 6A, lower heatmap), we focused on SMC6 because the increase was statistically significant. The imaging for SMC5 did not show a definite phenotype, reinforcing the lack of statistical significance of the iPOND fold change. To address the reviewer's query, we have now included the SMC5 staining (Lines 362-363, Fig S4B). Since SMC5 and SMC6 function in a complex together, we hypothesize that loss of one or the other may impact the virus, despite the localization differences. Indeed, we see that loss of either SMC5 or SMC6 impacts viral DNA replication and progeny formation, likely due to the necessity of both proteins in the function of the complex. We have added further explanation of our rationale for testing siRNA knockdown of SMC5 and SMC6 in the Results section (Lines 369-371).

- Difficult to see the different lines in the different shades of grey. Maybe change colors.

>We have revised Figure 7 to make the differences in line color more pronounced.

- How can the different effects be explained for the three targets? Why does siSMARCA1 not increase viral genomes in $\Delta E4$? Please clarify.

>Upon individual depletion of either SMARCA1, SMC6, or SMC5, we observed modest inhibitory effects on both viral DNA replication and infectious progeny production. We describe in the Discussion possible explanations for our observations. For example, given that the proteins are each a part of families containing multiple proteins, which may coordinate with each other, these observations could be explained by functional redundancy within the host response that is more apparent during Ad $\Delta E4$ relative to Ad WT infection. It is possible that depletion of SMARCA1 only displayed a modest inhibitory effect on Ad WT but not Ad $\Delta E4$ viral DNA replication because of more apparent redundancy during Ad $\Delta E4$ relative to Ad WT infection. Therefore, depletion of only SMARCA1 may not be sufficient to overcome other deficiencies of the $\Delta E4$ virus. We have added further explanation of this observation in the Results section (Lines 379-385). Indeed, we observed that another SWI/SNF-related protein, SMARCA6/HELLS decreased in abundance in a E4-dependent manner, and SMARCA1 has been observed to be both targeted for degradation by Ad5 WT via an interaction with E1B55K and localized to VRCs (PMID 30996091).

Reviewer #3 (Comments for the Author):

The authors present a detailed analysis of the changing cellular proteome during adenovirus infection, and explore specifically the changes in protein association with the viral genome that arise as a result of deletion of the viral E4 gene, which is known to encode several proteins that target cellular proteins for degradation. Their overall conclusions are (1) that there are several kinetic classes of protein that are depleted from infected cells in an E4-dependant manner, and (2) that E4 proteins cause loss of proteins from the SWI/SNF and SMC families that otherwise associate with replicating viral DNA; they further show that these proteins are inhibitory to the productivity of infection. Overall, there is considerable novelty in the study, particularly including the use of iPOND to look at proteins associated with the viral DNA, which will be of substantial interest to fellow scientists in the virology community.

The manuscript is clearly written and the data presentation is of very high standard. The data support their conclusions, in most respects, very well. I list some points that need to be considered below.

Comments:

Fig 3A, B and Line 179-192: It seems logical to assume that the three temporal classes of cellular protein degradation targets have been made mutually exclusive, ie proteins that are reduced of mock at 16 hr are still reduced of the mock at 24 and 48 hr but they have been subtracted out from the proteins found to be reduced at these later time points to create the lists for GO analysis. Is that correct? If so, the text needs to be improved to make that clear.

>We thank the reviewer for pointing out this potential confusion. In fact, the GO analysis was performed for ALL proteins decreased at the respective time points. The analysis included those proteins that were decreased at the considered time point as well as those decreased at previous time points, as long as the protein was still significantly decreased at the considered time. The goal of the GO analysis was to investigate the cellular processes that are impacted by decreased host proteins over the course of the infection time course. We reasoned that a decreased protein at a specific time point will impact the cellular processes effected at that time point, even if that protein was decreased at an earlier time as well. Therefore, we considered all proteins decreased at the respective time points. As a technical matter, there were not enough proteins that were decreased exclusively at 16hpi (27 proteins) or 24hpi (96 proteins) (please refer to Figure 2C) to generate enriched terms for the GO analysis. We performed a GO analysis for the proteins exclusively decreased at 48hpi (1,485 proteins) and found very similar results to the analysis performed for the manuscript and included in Figure 3B. Specifically, the “Chromosomal Organization” term, which we focus on for the experimental follow-up, has an FDR value of 8×10^{-18} so is indeed very significantly enriched. We have revised the text of the Results section (**Lines 174-183**) to clarify the inputs for the GO analysis.

Fig 4C, D and Line 237: This conclusion is hard to justify without detailed quantitation of the relevant protein bands, with normalization to the control, across multiple biologic repeats. By eye, any difference between wt and mutant in the extent of loss of the newly identified targets (RECQL4, SMARCA6, SQSTM1, DCAF1) seems minimal at best, especially when compared to the complete protection from degradation by each of the mutations of the well-established degradation targets RAD50 and MRE11. The authors do go on (line 240-3) to discuss this difference, but the primary conclusion seems overstated from the data. These are indeed E4-dependent degradation targets but presenting them primarily as targets of the E1B/E4 Orf6 Ub-ligase is too strong.

>The results shown for immunoblot analysis are representative of multiple biological replicates and recapitulate the abundance changes quantified in the mass spectrometry data. We have now

quantified the protein abundances using densitometry measurements (Figure S2) (Methods Lines 556). We agree that these proteins are not the sole targets of the E1B55K/E4orf6 ligase. We have now further clarified this in the Results text and highlighted the differences in abundance changes between the new targets (RECQL4, SMARCA6, SQSTM1, and DCAF1) and those that we observe for well-established substrates of the E1B55K-E4orf6 ligase (RAD50 and MRE11) (Lines 237-245).

Line 265 and following. It's unclear why the work switched from A549 (a well-established host system for Ad5) to U2OS cells. The latter are less infectable by the virus, which the authors clearly know as they increased the moi for the iPOND series of experiments in U2OS to 40 rather than the 10 used for the A549 work, which should achieve approx. similar levels of infection. The difference in cell line does though make bringing the two sets of data together rather difficult as the stage of infection at a given time p.i. cannot be assumed to be the same.

>The U2OS cell line was originally chosen for iPOND analysis so that data could be compared to other proteomics data involving DNA viruses. We have included a statement describing that rationale in the Methods section (Lines 599-601). However, when integrating the whole cell proteome data we chose to use A549 cells, which is a more well-established host system, as the reviewer points out. We recognize that this adds a complication when comparing datasets, however, we considered proteins that are dynamically altered in both the iPOND and whole cell proteome. We therefore feel that, despite the differences in infection kinetics, virus-dependent alterations of the host proteome can be detected in the data. The experimental follow-up of selected targets (immunoblot analysis showing protein abundance, immunofluorescence showing localization, and siRNA knockdown to measure impact on viral replication), are all performed in the more relevant A549 cells.

Fig 5C and Fig S2D, line 290-307. The replicating DNA that is the source of the protein for MS proteomics will be a mix of viral and cell DNA. In order to draw the conclusions that are offered about differences in specific protein enrichment between Ad5wt and Δ E4, it needs to be demonstrated that the proportion of the labelled (replicating) DNA that is viral is similar in each case. Otherwise a difference between the two could be down to a greater abundance of viral DNA in one set of samples.

>The reviewer raises an important point about the relative amounts of viral DNA measured in the iPOND, which could be impacted by differences in replication kinetics of the Ad5 WT and Δ E4 viruses. We feel that our data suggest that differences in viral DNA are not generally driving our iPOND abundance quantifications toward artifactually identifying enrichment on the Ad5 wildtype or mutant genomes. Evidence for this hypothesis can be provided by the very similar amounts of enriched proteins identified on the Ad5 wildtype (236 proteins) or Δ E4 (264 proteins) genomes (Figure 5B), which would not be expected if differences in viral DNA amounts were driving differences in enrichments. This balance between protein enrichments on each genome can also be observed, qualitatively, in the volcano plot comparing the fold changes and statistics for Ad wildtype and Δ E4 abundances (Figure 5C). Finally, if the Δ E4 mutant did significantly impact the proportion of labeled viral DNA, it would be expected to be associated with a decrease in viral DNA compared to wildtype infection. Since we are primarily interested in the proteins that are enriched on the Δ E4 genome in our data we expect that any potential negative impact on replication for the Δ E4 virus would only strengthen the confidence in the data which show proteins significantly increased on the Δ E4 viral genomes. We agree with the reviewer that this is a very important consideration as there may be differences in the iPOND data when comparing WT or

mutant infections. To further address this, we believe our follow-up experimentation using orthogonal techniques validated the iPOND targets that we identified.

Line 603-5. How were the data from each of the four gel slices from a sample handled subsequent to collection? At what point, if at all, were they brought back together? What happens to a protein that is in fraction 2 from one replicate and - being at the margin - ends up in fraction 3 from the other replicates? Given the declared aim of the fractionation is to improve discovery rates, the measured abundance of a protein might change simply for moving from one fraction to another. Also, how was the quantitation normalized across the four fractions taken from a sample?

>We apologize for being unclear. Gel fractions were run separately in the mass spectrometer and then combined at the data analysis step, which is a common approach used to improve proteome coverage and false discovery rates. The Proteome Discoverer software handles fractions of the same sample as a single run. This way, if peptides from the same protein are identified in different fractions (although unlikely), they are merged to estimate the protein abundance as if they were from the same run. Quantification was performed using a label-free approach using the "Precursor ions quantifier" node, and peptide abundances were rolled up into protein abundance using the summed abundance algorithm using only unique or razor peptides. We have specified this in text of the Methods section (Lines 646-651).

Line 670. Were the data from the four peptide bins for each sample combined together before doing the protein discovery analysis? If not, differences at the fractionation margins might give rise to a peptide 'moving' between fractions and so not giving a positive detection in the required 3 of 4 replicates.

>We believe the previous answer addresses the reviewer's concern about how data files from separate runs on the mass spectrometer can be analyzed as a single sample by Proteome Discoverer. We used Proteome Discoverer to merge fractions of the same sample as "fractions", therefore not treating them as different samples. Thus, if peptides from the same protein are identified in different fractions, it will still be identified in the respective sample from which the fraction came. Additionally, we wish to clarify that the sample fractionation for mass spectrometry analysis is not equivalent to the biological replicates assessed for the proteomics data analysis. The sample fractionation for mass spectrometry analysis was performed specifically for each of the four biological replicates. Therefore, any differences at margins of the fractions within a biological replicate would ultimately still be reflected as identified and quantified in that biological replicate. For the proteomics data analysis, a positive identification was made on the level quantification within the biological replicate (regardless of the fraction within the replicate). Therefore, the fraction in which a protein was measured by mass spectrometry does not impact the number of biological replicates in which the proteins was identified. We have clarified the text in the Methods section (Lines 688-694).

Some of the cited refs are not complete (at a glance, this includes 83, 84, 85 and 87).

>We thank the reviewer for pointing out the incomplete references, which were the result of the removal of some bibliographic information during the automatic uploads of citations from PubMed to Endnote. We have reviewed the list of references and corrected those were incomplete. In addition to the citations pointed out by the reviewer, we have changed others to match what the respective journal's website suggested for reference information:

Jean Beltran PM et. al. 2017

Pancholi NJ et. al. 2017

Atkins A et. al. 2020

Rother MB et. al. 2017
Nazeer R et. al. 2019
Abdul F et. al. 2018
Bentley P et. al. 2018
Dupont L et. al. 2021
Xu W et. al. 2018
Allweiss L et. al. 2021
Saribas S et. al. 2020
Niu C et. al. 2017

August 9, 2021

Dr. Matthew D. Weitzman
University of Pennsylvania Perelman School of Medicine
Department of Pathology and Laboratory Medicine
4050 Colket Translational Research Building
3501 Civic Center Blvd
Philadelphia, PA 19104-4318

Re: mSystems00468-21R1 (Adenovirus remodeling of the host proteome and of host factors associating with viral genomes)

Dear Dr. Weitzman:

Your manuscript has been accepted, and I am forwarding it to the ASM Journals Department for publication. For your reference, ASM Journals' address is given below. Before it can be scheduled for publication, your manuscript will be checked by the mSystems senior production editor, Ellie Ghatineh, to make sure that all elements meet the technical requirements for publication. She will contact you if anything needs to be revised before copyediting and production can begin. Otherwise, you will be notified when your proofs are ready to be viewed.

Publication Fees:

We recognize that the video files can become quite large, and so to avoid quality loss ASM suggests sending the video file via <https://www.wetransfer.com/>. When you have a final version of the video and the still ready to share, please send it to Ellie Ghatineh at eghatineh@asmusa.org.

Sincerely,

Jack Gilbert
Editor, mSystems

Journals Department
Fig S6: Accept
Table S2: Accept
Fig S5: Accept
Fig S3: Accept
Fig S2: Accept
Table S1: Accept
Fig S4: Accept
Fig S1: Accept